# DPP8/9 processing of human AK2 unmasks an IAP binding motif

Kim J Lapacz [ID][1,6], Konstantin Weiss [ID][1,6], Franziska Mueller [ID][2,3,6], Yuxing Xue[3], Simon Poepsel [ID][4,5], Matthias Weith [ID][1], Tanja Bange [ID][3✉] & Jan Riemer [ID][1,5✉]

## Abstract

**Adenylate kinase 2 (AK2) is localized in the intermembrane space of mitochondria, where it ensures efficient adenine nucleotide exchange between cytosol and mitochondria. For mitochondrial import, AK2 relies on the MIA40 disulphide relay system. Its cytosolic stability is subject to regulation through N-terminal processing by the dipeptidyl peptidases DPP8 and DPP9, which sensitize AK2 for proteasomal degradation. Here, we find that cytosolic AK2 degradation is mediated by Inhibitors of Apoptosis (IAPs), a class of E3 ligases that interacts with target proteins by binding to IAP-binding motifs (IBM). We have identified an IBM at the very end of AK2's novel N-terminus, which becomes exposed due to processing by DPP8/9. N-terminal acetylation mediated by the N-acetyltransferase NatA prevents this AK2-IAP interaction, therefore stabilizing AK2 in the cytosol. Performing a genome-wide in silico screen, we could identify 129 potential substrates in which an IBM becomes potentially unmasked by DPP8/9 processing. For one of these potential substrates, EIF2A, we demonstrate its targeting to IAPs after IBM exposure by DPP8/9 indicating that DPP8/9-mediated unmasking of IBMs is a general phenomenon.**

**Keywords** AK2; DPP9; Inhibitor of Apoptosis (IAP) Proteins; N-terminal Acetylation; IAP-binding Motif
**Subject Categories** Organelles; Post-translational Modifications & Proteolysis

## Introduction

When a nascent polypeptide chain exits the ribosome, it undergoes multiple processing steps for maturation. Among them are the removal of the N-terminal initiator methionine by aminopeptidases (MAP) (Giglione et al, 2004), acetylation by N-terminal acetyl transferases (NATs) (Deng and Marmorstein, 2021), and further processing steps by different peptidases. These processes generate mature N-termini thereby ensuring correct folding, proper localization and modulating the stability of the protein in the cytosol (Pfeffer et al, 2017; Shimshon et al, 2024; Smalinskaite et al, 2022; Voorhees et al, 2014; Voorhees and Hegde, 2015; Wang et al, 2021; Zhang et al, 2020).

Interestingly, about 5% of all human proteins expose so-called "inhibitor of apoptosis binding motifs" (IBM) at their N-terminus after MAP cleavage (Mueller et al, 2021). These four amino acid motifs enable proteins to bind and activate ubiquitin ligases known as inhibitors of apoptosis proteins (IAPs) leading to the degradation of the IBM-exposing protein (Silke and Meier, 2013; Vucic et al, 2011). In this context, IAPs bind to caspases through IBM motifs and thereby suppress apoptosis (Berthelet and Dubrez, 2013; Silke and Meier, 2013). IBMs, however, are only functional when they are exposed and are present unmodified at the very N-terminus of a protein. Thus, to protect cytosolic IBM-containing proteins from constant degradation but also from aberrantly activating apoptosis by replacing caspases from IAPs, most of them are N-terminally acetylated and thereby lose the ability to bind to IAPs (Mueller et al, 2021).

IBMs can also be hidden within proteins and only become exposed upon further processing. This is the case for two proteins that reside in the intermembrane space of mitochondria (IMS), SMAC/DIABLO and HtrA2 (Burri et al, 2005; van Loo et al, 2002). These two proteins are synthesized with a so-called bipartite mitochondrial targeting sequence (bpMTS). The bpMTS guides SMAC/DIABLO and HtrA2 into the IMS and becomes subsequently proteolytically removed (Burri et al, 2005; van Loo et al, 2002). bpMTS-removal exposes at their N-terminus an IBM which is "inert" as long as both proteins reside in the IMS. Upon apoptosis induction, both proteins are released to the cytosol, interact with IAPs and antagonize their anti-apoptotic function (Liu et al, 2000; Martins et al, 2002; Muñoz-Pinedo et al, 2006; van Loo et al, 2002; Verhagen et al, 2002; Wu et al, 2000). Thus, the bpMTS serves as an elegant way to mask the IBM in SMAC/DIABLO and HtrA2 en route to mitochondria after their cytosolic synthesis.

Another IMS protein that undergoes a complex processing pathway en route to mitochondria is adenylate kinase 2 (AK2). Conversely to SMAC/DIABLO or HtrA2, AK2 does not contain an N-terminal bpMTS (Finger et al, 2020; Petrungaro et al, 2015), but

[1]Redox Metabolism Group, Institute for Biochemistry, University of Cologne, 50674 Cologne, Germany. [2]Department of Mechanistic Cell Biology, Max Planck Institute of Molecular Physiology, 44227 Dortmund, Germany. [3]Institute of Medical Psychology and Biomedical Center (BMC), Faculty of Medicine, LMU Munich, 80336 Munich, Germany. [4]Center for Molecular Medicine Cologne (CMMC), Faculty of Medicine and University Hospital, University of Cologne, 50931 Cologne, Germany. [5]Cologne Excellence Cluster on Cellular Stress Responses in Aging-Associated Diseases (CECAD), University of Cologne, 50931 Cologne, Germany. [6]These authors contributed equally: Kim J Lapacz, Konstantin Weiss, Franziska Mueller. ✉E-mail: tanja.bange@med.uni-muenchen.de; jan.riemer@uni-koeln.de

instead conserved cysteines that enable its import into the IMS by mitochondrial disulphide relay (Backes and Herrmann, 2017; Chacinska et al, 2009; Edwards et al, 2020; Habich et al, 2019) (Finger et al, 2020). After translation, AK2 is processed by methionine aminopeptidases (MAP, removal of the N-terminal "M") and the dipeptidyl peptidases DPP8 and DPP9 (DPP8/9, removal of the N-terminal dipeptide "AP") (Finger et al, 2020). The latter processing exposes a neo-N-terminus that destabilizes AK2 in the cytosol and primes it for proteasomal degradation. Thereby, for AK2, proteasomal degradation and mitochondrial import compete as the ubiquitin-proteasome system cannot act on AK2 as soon as it reaches the IMS. Consequently, preventing AK2 import, e.g., by mutating the cysteines required for import leaves AK2 in the cytosol exposed to proteasomal degradation and destabilizes it. Conversely, inhibition of DPP8/9 processing, mutations in the DPP8/9 cleavage site or changes of the neo-N-terminus to more stabilizing residues stabilize AK2. This does not only increase levels of AK2 in mitochondria but also leads to the accumulation of AK2 in the cytosol. Notably, since AK2 lacks a cleavable bpMTS, any cytosolic modifications at its N-terminus remain present also in the IMS (Finger et al, 2020).

In this study, we uncovered a new mode of IBM exposure upon proteolytic processing. We found that DPP8/9 processing reveals a previously masked IBM at the N-terminus of AK2. This IBM allows AK2 to be targeted by the IAPs BIRC2, BIRC3, BIRC6 and XIAP, and explains the rapid degradation of the protein in the cytosol. Acetylation of MAP- and DPP8/9-processed AK2 masks the IBM again and contributes to AK2 stability. This mechanism might also hint at a regulatory pathway by which low DPP8/9 activity would leave the IBM in AK2 hidden and inert. AK2 is one of many proteins, in which an IBM is hidden by a DPP8/9-processing site—in a bioinformatics screen we identified a total of 129 possible candidates in the human proteome. For one of those candidates, the eukaryotic translation initiation factor EIF2A, we demonstrate that it is targeted by IAPs after DPP8/9-dependent processing in intact cells.

# Results and discussion

## The majority of cellular AK2 is processed by cytosolic DPP8/9

AK2 is an IMS protein that en route to mitochondria is modified at its N-terminus at least three times (Fig. 1A); it undergoes cleavage by MAP, removing the initiator N-terminal methionine, it becomes processed by DPP8 and 9 removing the dipeptide "AP" from its N-terminus after MAP cleavage, and lastly, it becomes N-terminally acetylated (Finger et al, 2020; Mueller et al, 2021; Wilson et al, 2013). This N-terminal acetylation has been identified on tryptic peptides in proteomics starting with "$NH_2$-SV..." indicating that it can occur after DPP8/9 processing (Finger et al, 2020; Mueller et al, 2021) (Fig. 1A).

We previously demonstrated that processing by DPP8/9 targets AK2 for proteasomal degradation, and that the competing mitochondrial import protects AK2 as it becomes inaccessible to the ubiquitin-proteasome system (Finger et al, 2020). Using an AK2 variant that cannot be imported into mitochondria (AK2-3CS) (Finger et al, 2020), we confirmed that AK2 levels could be

increased by proteasomal inhibition (Fig. 1B), and by DPP9 knockout (Fig. 1C,D). We assume that DPP8/9 processing leads to efficient removal of cytosolic AK2 ensuring its exclusive mitochondrial localization despite its import being driven by a slow import machinery (Finger et al, 2020). In line, in DPP9 knockout, endogenous AK2 accumulated in the cytosol (Fig. 1E), presumably because it began folding before translocation into mitochondria.

The extent to which DPP8/9 processes the AK2 N-terminus remains unclear. We thus compared the presence and abundance of N-terminal AK2 peptides in WT and DPP9 KO cells using mass spectrometry (Fig. 1F; Appendix Fig. S1). We thereby identified three different N-terminal peptides of AK2, $NH_2$-SVPAAEPEYPK (processed by MAP and DPP8/9), $NH_2$-APSVPAAEPEYPK (processed by MAP) and acetylated $NH_2$-MAPSVPAAEPEYPK. We did not identify further acetylated N-termini. When we analyzed the raw intensities of the three N-terminal peptides of AK2, we found that the amount of the MAP and DPP8/9-processed N-terminal peptide was decreased by two third in DPP9 KO cells. Conversely, the peptide amounts of the MAP-processed and the acetylated N-terminal peptides increased. To complement these data, we developed a cysteine-based shift assay to assess the extent of DPP8/9 processing of AK2 (Appendix Fig. S2). We generated an AK2 variant in which we replaced the alanine at position 2 by a cysteine residue (A2C), which has previously been shown to not influence MAP or DPP8/9 cleavage efficiency in vitro (Geiss-Friedlander et al, 2009; Xiao et al, 2010). This cysteine can only be modified if DPP8/9 cleavage did not happen. We thereby found in unperturbed HEK293 cells, 80% of cellular AK2 is present in the DPP8/9-processed state.

We conclude that multiple processing steps affect AK2 and that the majority of cellular AK2 becomes processed by DPP8/9 indicating that the prevalent N-terminus of AK2 is "$NH_2$-SV...".

## DPP8/9 processing of human AK2 unmasks an IAP-binding motif

A closer look at N-terminal amino acids of AK2 revealed that DPP8/9 processing exposes a potential IBM, $NH_2$-SVPA-.... IBMs comprise a four-residue, loose consensus sequence (Eckelman et al, 2008; Shi, 2002) (Fig. 2A) that binds and activates IAPs. Interestingly, prominent proteins containing IBMs such as SMAC/DIABLO (IBM: $NH_2$-AVPI-...) and HtrA2 (IBM: $NH_2$-AVPS-...) reside in the IMS. Indeed, a structure prediction using Alphafold 3 (Abramson et al, 2024) showed binding of the AK2 "SVPA"-N-terminal motif to the BIR3 domain of XIAP with high confidence, consistent with the reported structure of the similar "AVPI" tetrapeptide of the N-terminus of SMAC/DIABLO bound to the XIAP BIR3 domain (Wu et al, 2000) (Fig. 2B). Systematic studies have shown that "SVPI" tetrapeptides can bind either the BIR2 and BIR3 domains with a preference for the BIR3 domain, although with slightly lower affinity than "AVPI" tetrapeptides (Lukacs et al, 2013). At the P4 position of the peptide, alanine is tolerated similarly to isoleucine (Lukacs et al, 2013; Sweeney et al, 2006), supporting the binding capability of the processed AK2 N-terminus. In addition, the N-terminal 15 amino acids of AK2 are predicted to be disordered and exposed, granting accessibility of folded AK2 to IAPs (Fig. 2C).

We next experimentally tested whether AK2 could bind to IAPs. To this end, we performed peptide pull-down experiments. We first

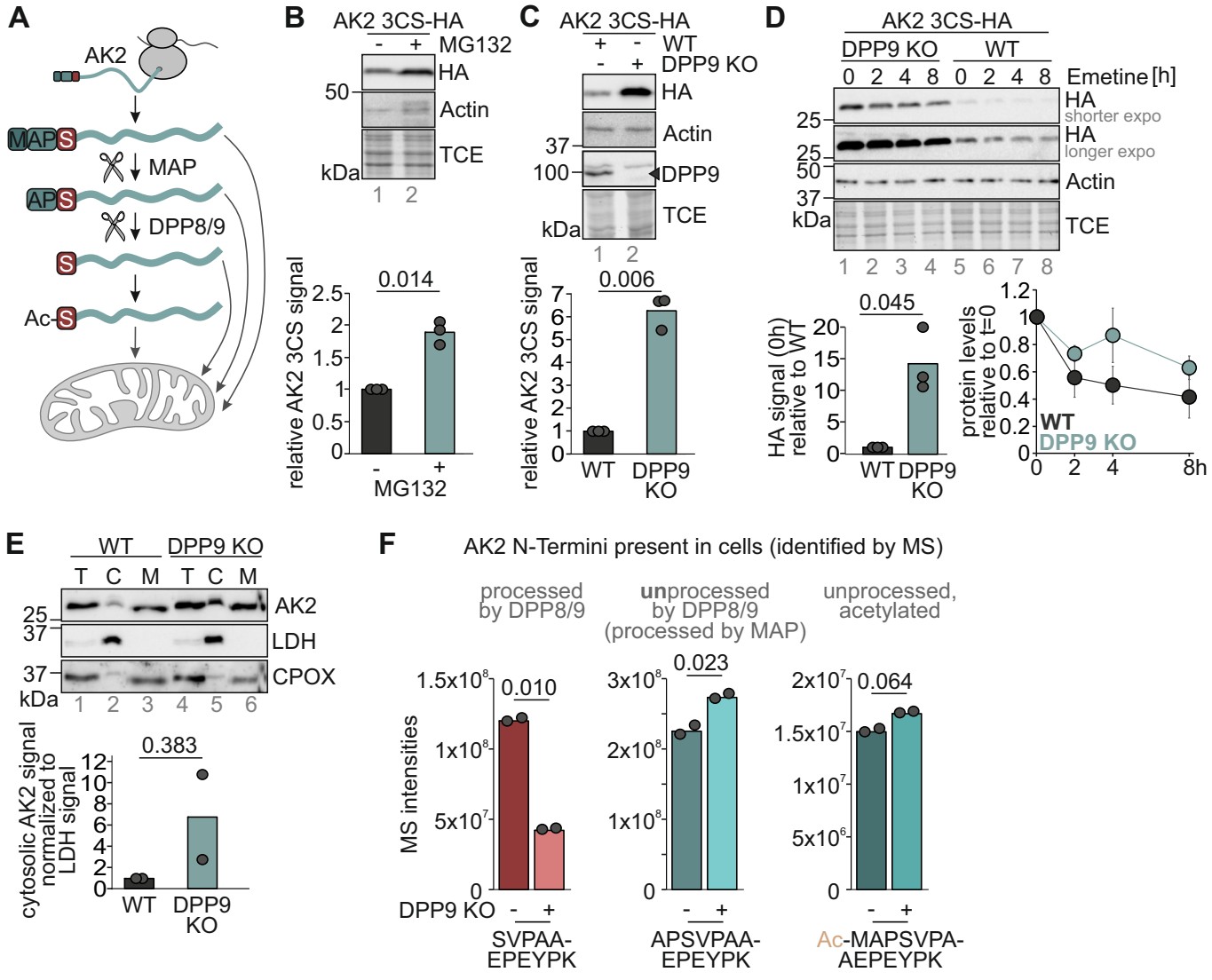

**Figure 1. The dipeptidyl peptidases DPP8/9 process the majority of AK2.**

(A) AK2 is expressed in the cytosol. Its initial methionine is cleaved off by MAP. Afterwards, DPP8 and DPP9 can process AK2 at its N-terminus removing two further amino acid residues. Both, processed and unprocessed AK2 can be acetylated at the N-terminus. All AK2 variants can be imported into the IMS of mitochondria. (B) Cytosolic AK2 is degraded by the proteasome. A stable inducible HEK293 cell line containing the import-incompetent AK2 variant AK2 C40,42,92S-HA was treated with the proteasome inhibitor MG132 for 6 h. Samples were lysed and analyzed via SDS-PAGE and immunoblot. $n = 3$ biological replicates. (C) Loss of DPP9 increases levels of cytosolic AK2. Stable inducible HEK293 cell lines containing AK2 C40,42,92S-HA in either WT or DPP9 KO background were lysed and analyzed by SDS-PAGE and immunoblot. $n = 3$ biological replicates. (D) Loss of DPP9 stabilizes cytosolic AK2. Stable inducible HEK293 cell lines containing AK2 C40,42,92S-HA in either WT or DPP9 KO background were tested in a chase experiment using the translation inhibitor emetine for the indicated times. Samples were lysed and analyzed by SDS-PAGE and immunoblot. Error bars represent standard deviation of three biological replicates. $n = 3$ biological replicates. (E) AK2 accumulates in the cytosol upon loss of DPP9. WT and DPP9 KO HEK293 cells were fractionated. The resulting fractions (T total, C cytosol, M mitochondria) were analyzed via SDS-PAGE and immunoblot. The cytosolic HA signal was normalized for the LDH signal. $n = 2$ biological replicates. (F) In intact cells, three different N-terminal peptides of AK2, NH2-SVPAAEPEYPK, NH2-APSVPAAEPEYPK and Ac-NH2-MAPSVPAAEPEYPK, were identified by MS analysis. The amounts of the NH2-SVPAAEPEYPK strongly depends on the presence of DPP9. HEK293 WT and DPP9 KO HEK293 cells were lysed and fractionated by high-pH reverse phase (9 fractions). We quantified 13,262 proteins in this dataset. The raw intensities of the three N-terminal peptides of AK2 comparing HEK293 WT with DPP8/9 KO cells are plotted as bar charts for each peptide. For "peptides w/o N-terminus" mean intensities of internal non-N-terminal peptides (WT $n = 40$; KO $n = 39$) were calculated and plotted. $n = 2$ biological replicates. Source data are available online for this figure.

synthesized biotinylated peptides identical in sequence to the N-terminus of AK2 in its unprocessed or processed forms (NH2-MAPSVPAAEPEYPKG…, Fig. 2D). These peptides represented the N-terminus existing after DPP8/9 cleavage (NH2-SVPAAEPEYP-KK-Biotin), the N-terminus as it would be present without DPP8/9

cleavage (NH2-APSVPAAEPE-KK-Biotin), the N-terminus as it would be present without MAP processing (NH2-MAPSVPAAEP-KK-Biotin), and a modified N-terminus in which the first residue of the IBM, serine, was mutated to valine (NH2-VVPAAEPEYP-KK-Biotin). The corresponding AK2 variant (AK2-S4V) was stabilized

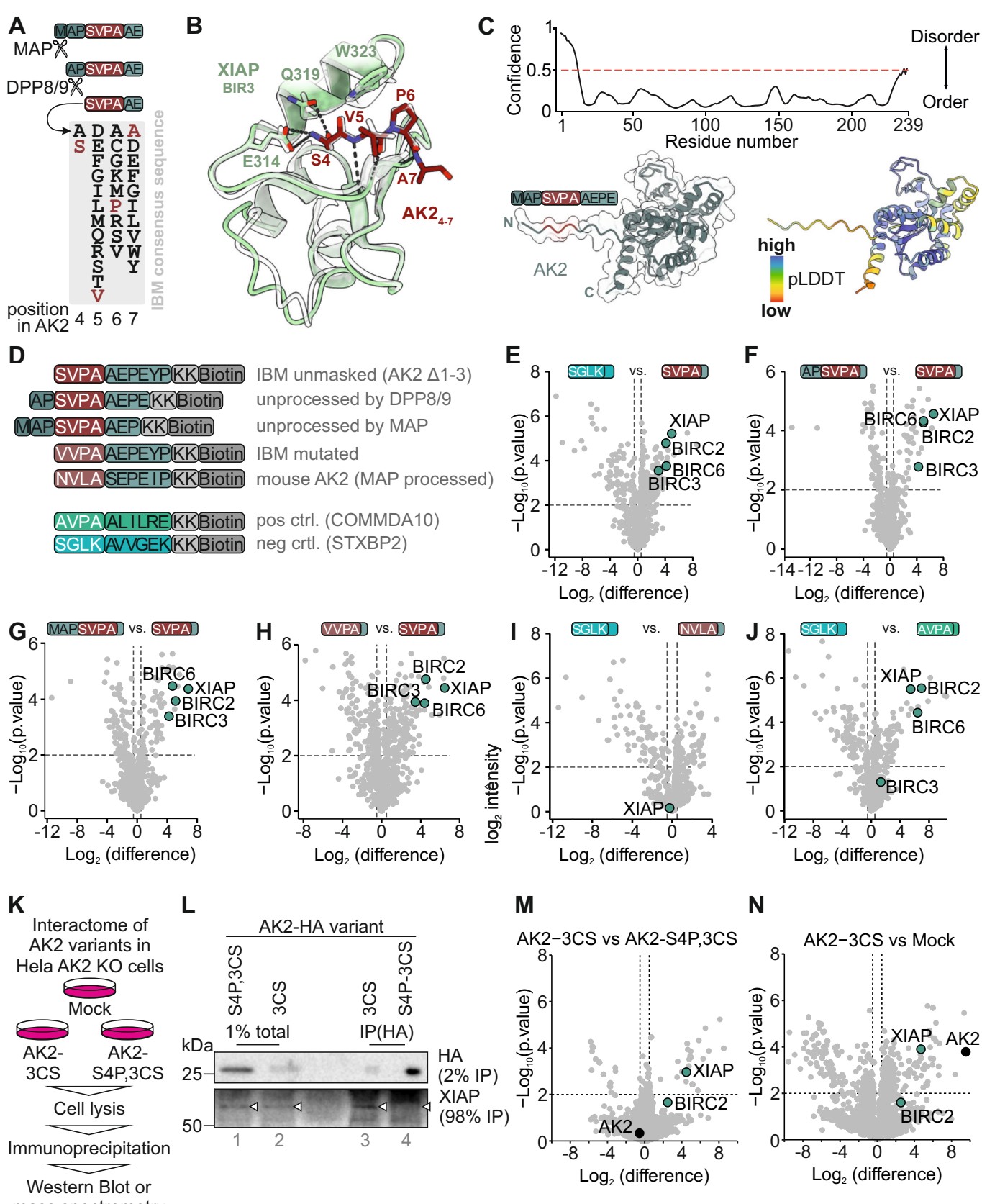

**Figure 2. Processing of AK2 by DPP8/9 unmasks an IBM at the N-terminus of AK2.**

(A) Processing of AK2 by MAP and DPP8/9 unmasks a putative IBM. The amino acid residues 4–7 of AK2 are "SVPA". They fulfill the loose requirement for an IBM. (B) Model of the processed N-terminus of AK2 (aa 4–7, dark red) bound to the BIR3 domain of human XIAP (aa 265–330, green). Binding of the N-terminal amino group is stabilized by electrostatic interactions between the N-terminus and XIAP E314, which would be disrupted by N-terminal acetylation. The interaction is further stabilized by H-bonds between the hydroxyl group of AK2 S4 and XIAP Q319, as well as backbone H-bonds between AK2 V5 and XIAP T308. The prediction aligns closely with the binding of an AVPA tetrapeptide bound to the XIAP BIR3 domain solved by X-ray crystallography (shown in light gray transparent, pdb 1G73 (Wu et al, 2000)). H-bonds and salt bridges shown as dashed black lines. (C) The N-terminal region of AK2 is disordered and exposed. *Top:* disorder prediction of full-length human AK2, showing a disordered region for the N-terminal 10 residues. Red dashed line indicates the 0.5 confidence cut-off to classify regions as disordered (Ishida and Kinoshita, 2007). Bottom left: alphafold model of human AK2, shown in ribbon and transparent surface representation. Bottom right: alphafold model of human AK2 colored according to the pLDDT score of alphafold. (D) Peptide constructs used for interaction screen. Various peptides containing amino acid residues representing different AK2 termini were used for an interaction screen using HeLa cell lysate. As a negative control the N-terminus of STXBP2 was used (light blue), as a positive control COMMDA10 (green). (E) The IBM motif of AK2 interacts with IAPs, but not the negative control of STXBP2. Peptides with the respective N-terminal peptide were incubated with HeLa cell lysates. Interaction partners were identified using MS. Pairwise, two-sided Student's *t* test was used for statistical testing. n = 3 biological replicates. (F) AK2 processed by MAP and DPP8/9 but not MAP-processed AK2 interacts with IAPs. Peptides with the respective N-terminal peptide of AK2 were incubated with HeLa cell lysates. Interaction partners were identified using MS. Pairwise, two-sided Student's *t* test was used for statistical testing. n = 3 biological replicates. (G) AK2 processed by MAP and DPP8/9 but not unprocessed AK2 interacts with IAPs. Peptides with the respective N-terminal peptide of AK2 were incubated with HeLa cell lysates. Interaction partners were identified using MS. Pairwise, two-sided Student's *t* test was used for statistical testing. n = 3 biological replicates. (H) Mutation within the IBM of AK2 abrogates the interaction with IAPs. Peptides with the respective N-terminal peptide of AK2 were incubated with HeLa cell lysates. Interaction partners were identified using MS. Pairwise, two-sided Student's *t* test was used for statistical testing. n = 3 biological replicates. (I) Neither the peptide representing the mouse AK2 N-terminus nor the negative control peptide of STXBP2 interacts with IAPs. Peptides with the respective N-terminal peptide of AK2 were incubated with HeLa cell lysates. Interaction partners were identified using MS. Pairwise, two-sided Student's *t* test was used for statistical testing. n = 3 biological replicates. (J) The positive control COMMDA10 reliably binds to various IAPs. Peptides with the respective N-terminal peptide of the controls were incubated with HeLa cell lysates. Interaction partners were identified using MS. Pairwise, two-sided Student's *t* test was used for statistical testing. n = 3 biological replicates. (K) Strategy to identify cytosolic interaction partners of AK2. The interactomes of the variants AK2 C40,42,92S-HA (3CS) and AK2 S4P, C40,42,92S-HA (S4P,3CS) were compared after immunoprecipitation by immunoblot or mass spectrometry. Both variants localize to the cytosol; however, the S4P variant is not processed by DPP8/9 and will thus not exhibit an IBM. (L) The IAP XIAP can bind to AK2. AK2 C40,42,92S-HA (3CS) and AK2 S4P, C40,42,92S-HA (S4P,3CS) were immunoprecipitated from HEK293 cells stably expressing both proteins. XIAP was detected using immunoblotting. AK2 exposing an IBM after DPP8/9 cleavage binds XIAP, while the S4P mutation prevents XIAP binding. (M) An N-terminal mutation of DPP8/9-processed AK2 (S4P) abolished binding of AK2 to XIAP and BIRC2. AK2 C40,42,92S-HA (3CS) and AK2 S4P, C40,42,92S-HA (S4P,3CS) were immunoprecipitated and binding partners were quantified by MS. In all, 2558 proteins were quantified, number of replicates n = 3. Data were imputed at the lower end of the distribution (downshift: 2; width:0.3). AK2 and IAPs are highlighted. (N) AK2 binds to XIAP and BIRC2 in the cytosol. AK2 C40,42,92S-HA (3CS) and mock-treated cells were immunoprecipitated and binding partners were quantified by MS. In total, 2623 proteins were quantified, number of replicates n = 3. Data were imputed at the lower end of the distribution (downshift: 2; width:0.3). AK2 and IAPs are highlighted. Source data are available online for this figure.

in cells despite being processed by DPP8/9, indicating that the subsequent degradation pathway is impaired (Finger et al, 2020). We also included a peptide representing the N-terminus of mouse AK2 as it would exist after DPP8/9 cleavage (NH2-NVLASEPEIP-KK-Biotin). Interestingly, mouse AK2 does not contain a hidden IBM. As a control, we chose the peptide NH2-SGLKAVVGEK-KK-Biotin (negative control). It derives from the protein STXBP2. After translation, the N-terminus of STXBP2 is NH2-MAPSGLK (the first four amino acids identical to AK2); after MAP and DPP8/9 processing, NH2-SGLK… would remain at the STXBP2 N-terminus - a sequence that does not adhere to the consensus of IBMs (Fig. 2A). We also included the peptide NH2-AVPAALILRE-KK-Biotin (positive control). It derives from the protein COMMDA10, which has previously been shown to efficiently precipitate IAPs (Mueller et al, 2021). Isolation of streptavidin-bound peptides after incubation in HeLa lysates and identification of interaction partners by mass spectrometry revealed that NH2-SVPAAEPEYP-KK-Biotin efficiently coprecipitated the IAPs BIRC2, BIRC3, BIRC6 and XIAP compared to the negative control NH2-SGLKAVVGEK-KK-Biotin (Fig. 2E). Likewise, these IAPs were significantly enriched with NH2-SVPAAEPEYP-KK-Biotin when compared to NH2-APSVPAAEPE-KK-Biotin (Fig. 2F), NH2-MAPSVPAAEP-KK-Biotin (Fig. 2G), and NH2-VVPAAEPEYP-KK-Biotin (Fig. 2H). The peptide representing the mouse AK2 N-terminus NH2-NVLASEPEIP-KK-Biotin did not enrich IAPs compared to the negative control NH2-SGLKAVVGEK-KK-Biotin (Fig. 2I), while the peptide representing the positive control (NH2-AVPAALILRE-KK-Biotin) faithfully coprecipitated IAPs compared to the negative control (Fig. 2J). Thus, only a peptide representing the MAP and

DPP8/9-processed N-terminus of human AK2 precipitates different IAPs, while mouse AK2 or not fully processed N-termini of human AK2 do not. Of note, the peptide representing the mouse N-terminus bound significantly to UBR5, which is an E3 ligase involved in the Arg/N-degron pathway (Shimshon et al, 2024; Tasaki et al, 2005).

We then aimed to verify the interaction between full-length AK2 and one of the IAPs in cell culture. This experiment is complicated by the fact that wildtype AK2 becomes rapidly degraded after DPP8/9 processing (which unmasks the IBM), and thus coprecipitation of IAPs with wildtype AK2 was not successful as only small fractions of this protein reside in the cytosol at any given time (Finger et al, 2020). We thus employed the cytosolic AK2 C40,42,92S-HA variant and compared its interactome to the AK2 S4P, C40,42,92S-HA variant by immunoblotting and mass spectrometry (Fig. 2K). The latter variant cannot be cleaved anymore by DPP8/9 and will not exhibit an IBM (Finger et al, 2020). Consequently, we immunoprecipitated XIAP only with AK2 C40,42,92S-HA but not with the S4P variant (Fig. 2L). We then performed mass spectrometric analysis on the HA-immunoprecipitates of cells expressing AK2 C40,42,92S-HA, AK2 S4P, C40,42,92S-HA or an empty vector (Fig. 2M,N; Appendix Fig. S3). Indeed, XIAP and BIRC2 were strongly enriched on AK2 C40,42,92S-HA compared to AK2 S4P, C40,42,92S-HA (Fig. 2M) and to mock (Fig. 2N). We complemented this approach with another experiment in which we constructed an AK2 variant that lacked the first three amino acids (mimicking the N-Terminus after MAP and DPP8/9 cleavage) and was equipped with the SMAC bpMTS (SMAC$^{MTS}$) (Appendix Fig. S4). The SMAC$^{MTS}$ becomes

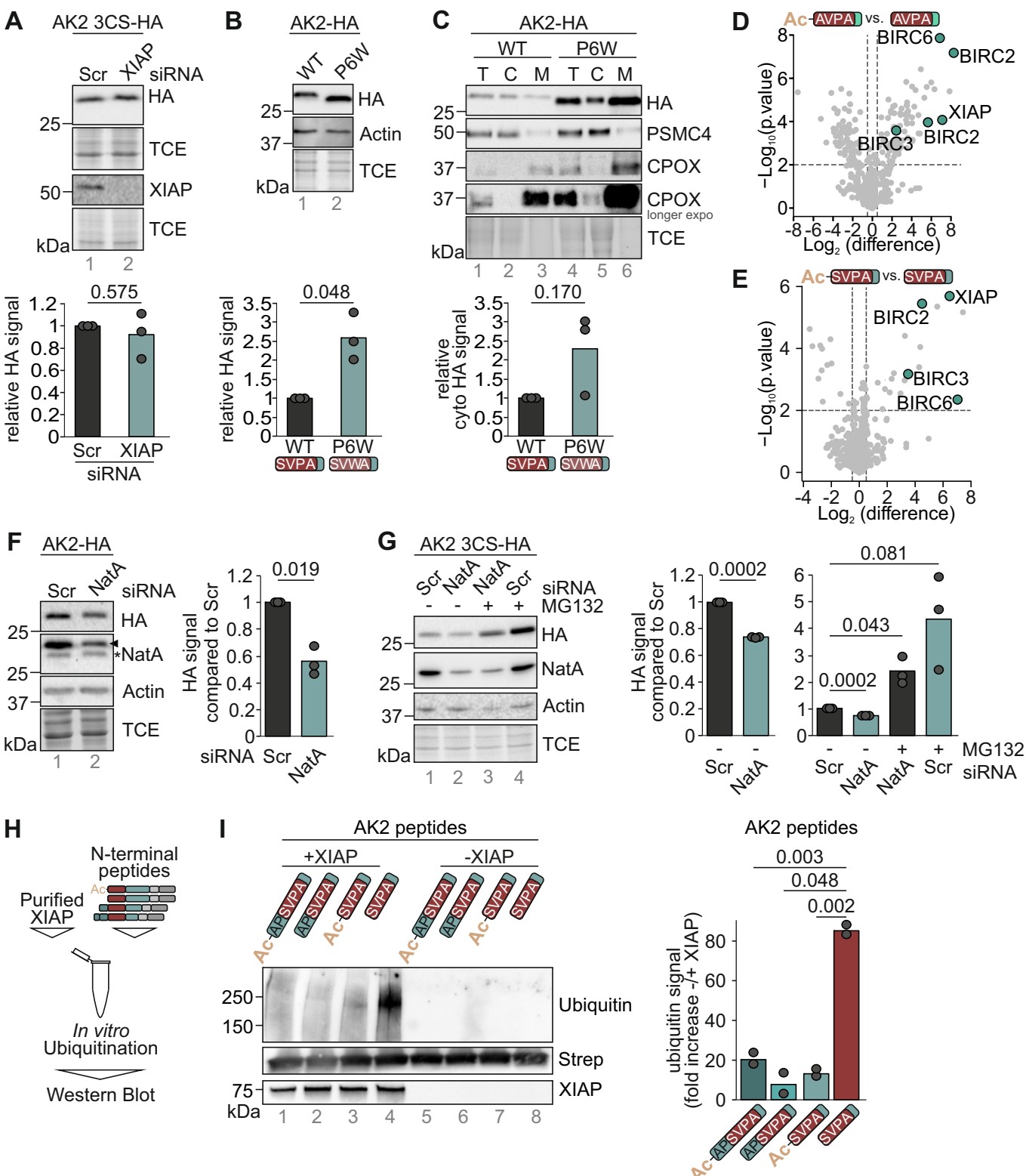

◄ **Figure 3.  Abrogation of the interaction of AK2 with IAPs prevents AK2 ubiquitination and stabilizes the protein.**

(A) HeLa cells expressing AK2 C40,42,92S in an AK2 KO background were seeded and XIAP levels were depleted using siRNA. Cell lysates were analyzed via SDS-PAGE and immunoblot. $n = 3$ biological replicates. (B) Mutation of the IBM of AK2 increases AK2 levels. Stable inducible cell lines expressing either the WT or the P6W variant of AK2 were lysed and analyzed via SDS-PAGE and immunoblot. $n = 3$ biological replicates. (C) IBM mutation leads to cytosolic accumulation of respective AK2 variant. Cells were fractionated and fractions (T total, C cytosol, M mitochondria) were analyzed via SDS-PAGE and immunoblot. Cytosolic HA signal was normalized for the PSMC4 signal. $n = 3$ biological replicates. (D) Acetylation of the IBM at the N-terminus of COMMDA10 prevents interaction with IAPs. Peptides with either the acetylated or the non-acetylated N-terminal peptide of COMMDA10 were incubated with HeLa cell lysates. Interaction partners were identified using MS. Pairwise, two-sided Student's $t$ test was used for statistical testing. $n = 3$ biological replicates. (E) Acetylation of the N-terminus of AK2 inhibits the interaction with IAPs. Either acetylated or non-acetylated N-terminal AK2 peptides were incubated with HeLa cell lysates. Interaction partners were identified using MS. Pairwise, two-sided Student's $t$ test was used for statistical testing. $n = 3$ biological replicates. (F) Depletion of NatA levels destabilizes AK2. HeLa expressing AK2-HA in an AK2 KO background were seeded and the levels of the catalytic domain of NatA were depleted using siRNA. Cell lysates were analyzed via SDS-PAGE and immunoblot. $n = 3$ biological replicates. (G) AK2 stability loss upon NatA depletion can be rescued by proteasomal inhibition. HeLa expressing AK2 C40,42,92S in an AK2 KO background were seeded and the levels of the catalytic domain of NatA were depleted using siRNA. 16 h before the experiment, the proteasome was inhibited by MG132. Cell lysates were analyzed via SDS-PAGE and immunoblot. $n = 3$ biological replicates. (H) Experimental scheme of the in vitro ubiquitination assay. Purified full-length GST-XIAP and biotinylated peptides representing different N-termini of AK2 were incubated and analyzed for ubiquitination using immunoblotting. (I) XIAP ubiquitinates the DPP8/9-processed, free N-terminus of AK2 in vitro. Different biotinylated peptides representing the N-terminus of AK2 ("MAP-cleaved" NH$_2$-APSVPAAEPE... in the acetylated or free form, and "DPP8/9 processed" NH$_2$-SVPAAEPEYP... in the acetylated or free form) were subjected to an in vitro ubiquitination assay using purified full-length GST-XIAP. An example of an immunoblot is shown on the left. On the right, the ratio of $-/+$ XIAP of quantified immunoblots ($n = 2$) for each peptide normalized to the streptavidin signal amounts is plotted. Source data are available online for this figure.

removed after import into the IMS, and since the precise cleavage site for the MTS is known (Du et al, 2000), we could construct the AK2 variant in a way that after cleavage a soluble AK2 exists in the IMS that harbors the IBM at its very N-terminus. In the IMS, this construct is hidden from cytosolic IAPs and protected from proteasomal degradation. Only upon complete native cell lysis, this AK2 variant can encounter cytosolic proteins. Indeed, when we immunoprecipitated this variant from the cell lysate we coprecipitated XIAP (Appendix Fig. S4C).

Our peptide pull-down and immunoprecipitation experiments thus demonstrated that the four amino acids of human AK2 present at its neo-N-terminus after DPP8/9 processing serve indeed as an IBM and can faithfully coprecipitate the IAPs BIRC2, BIRC3, BIRC6 and XIAP from HeLa lysate.

## Interfering with the IBM in AK2 prevents IAP binding and stabilizes cytosolic AK2

Next, we tested the relevance of the AK2 IBM and its crosstalk with N-terminal acetylation for the stability of human AK2 in cell culture. First, we depleted XIAP, which was the most prominent interaction partner of the IBM in AK2. We thereby observed no significant stabilization of cytosolically confined AK2-3CS (Fig. 3A). Thus, we next mutated the IBM in AK2. We set this mutation at position three of the IBM to ensure continued MAP and DPP8/9 processing. It has been shown that at the P3 position of IBMs (corresponding to AK2 P6), proline is conserved, while tryptophan is highly disfavored and disrupts IAP binding, in particular to BIR3 of XIAP (Lukacs et al, 2013; Sweeney et al, 2006). In agreement with this, an AK2-P6W variant was present at higher levels when compared to AK2 wildtype indicating a decreased degradation of the protein in the cytosol (Fig. 3B). In line with a decreased cytosolic degradation, AK2 P6W in part accumulated in the cytosolic fraction although mitochondrial levels increased even more (Fig. 3C). Likewise, another AK2 variant bearing the S4V mutation that can also not interact with IAPs ((Lukacs et al, 2013; Sweeney et al, 2006), Fig. 2H) was stabilized (Appendix Fig. S5 (Finger et al, 2020)), collectively demonstrating that an intact IBM determines AK2 stability in cells.

Next, we assessed how acetylation crosstalks with the presence of the IBM in human AK2. Recent work in human cells indicated a protective role of acetylation particularly to mask IBMs and thereby preventing targeting of the respective proteins to the E3 Ubiquitin ligases of the IAP family, which include XIAP, BIRC2 and BIRC3 (Mueller et al, 2021). Structurally, how the N-terminal acetylation would disfavor the engagement with IBM-binding domains can be rationalized e.g. for BIR3 of XIAP since the interaction of the N-terminal amino group with XIAP E314 would be disrupted. Using our peptide pull-down experiments, we tested for the effects of N-terminal acetylation, and compared peptides with and without N-terminal acetylation. We found that our positive control peptide NH$_2$-AVPAALILRE-KK-Biotin (COMMDA10) precipitated XIAP, BIRC2, BIRC3, and BIRC6, while its N-acetylated counterpart did not (Fig. 3D). Likewise, NH$_2$-SVPAAEPEYP-KK-Biotin (AK2 after DPP8/9 processing) precipitated XIAP, BIRC2, BIRC3, and BIRC6, and its N-acetylated counterpart did not (Fig. 3E). Next, we tested whether lowering the amount of N-acetylation in cells would affect AK2 levels. Of the seven N-terminal acetyl transferases in mammals only NatA appears to recognize peptide sequences starting with A or S, the N-terminal amino acids found in IBMs (Aksnes et al, 2019; Frottin et al, 2006). We therefore targeted NatA by siRNA-mediated knockdown of its catalytic NAA10 subunit, and analyzed resulting cell lysates by immunoblot. We thereby found a depletion of AK2 levels consistent with a role of N-acetylation in stabilizing AK2 (Fig. 3F). Acetylation prevented proteasomal degradation as MG132 treatment strongly stabilized AK2 in particular in siRNA NatA-treated cells (Fig. 3G).

Recognition of the AK2 IBM by e.g. XIAP should drive ubiquitination of the protein. We thus tested in an in vitro ubiquitination assay whether XIAP would be capable of ubiquitinating peptides representing different N-termini of AK2 (Fig. 3H). To this end, we incubated purified XIAP (Appendix Fig. S6) with peptides representing the DPP8/9-processed and unprocessed termini in their acetylated and non-acetylated forms. After incubation mixes were analyzed by immunoblotting against ubiquitination. We thereby found that only the NH$_2$-SVPA... peptide representing the free DPP8/9-processed AK2 N-terminus could be ubiquitinated above the background in the presence of XIAP (Fig. 3I, lane 4).

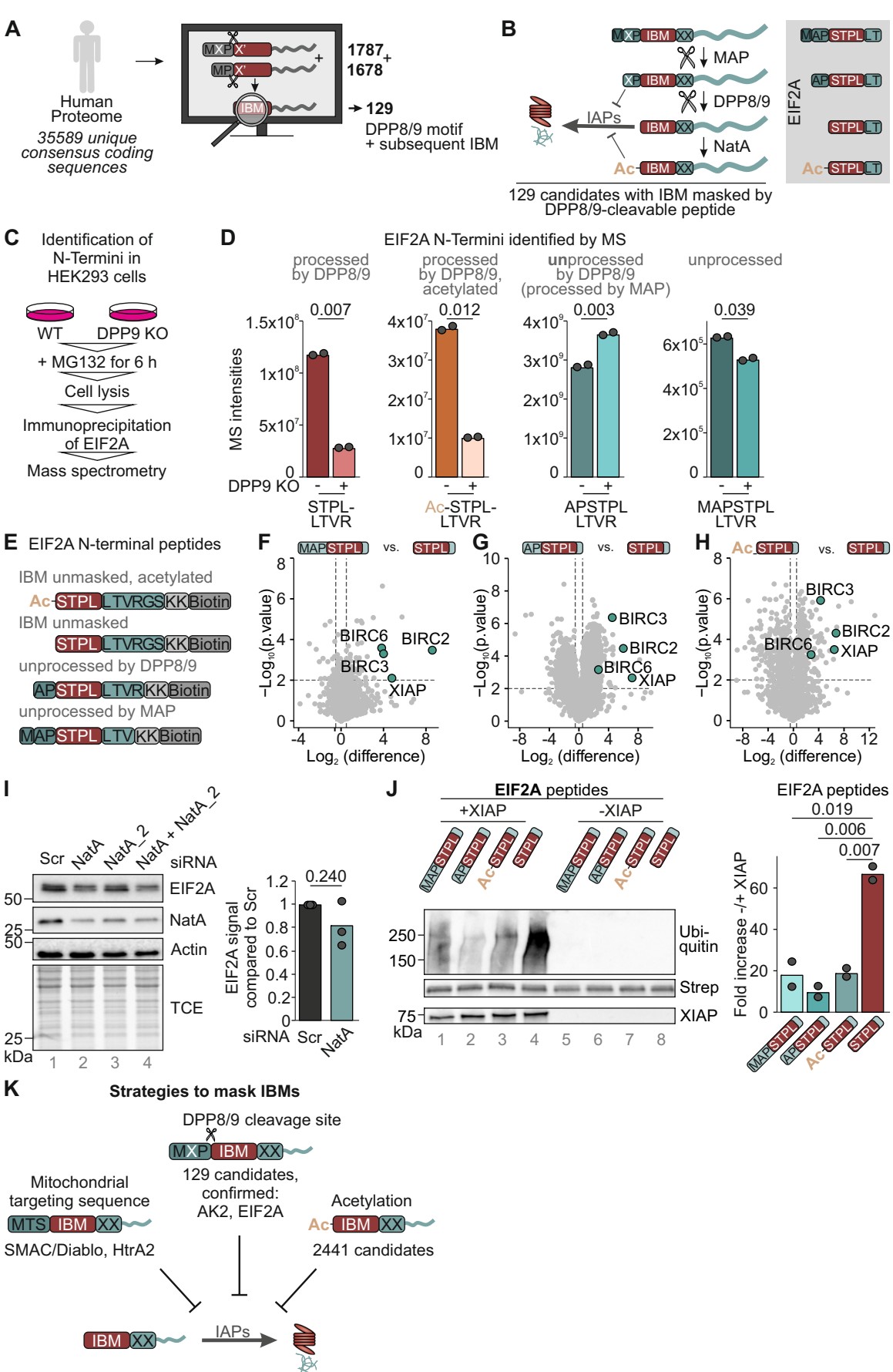

**Figure 4. A class of proteins contains IBMs masked by DPP8/DPP9-cleavable dipeptides.**

(A) A bioinformatic screen on human genome (35589 unique consensus coding sequences) was performed revealing 3465 proteins with potential DPP8/9 cleavage sites of which 129 contain an IBM behind the cleavage site (list, see Appendix Fig. S7). (B) Model of IBM unmasking by DPP8/9 processing. DPP8/9 processing results in the unmasking of an IBM which facilitates increased degradation of the protein. There appears to be an entire class of proteins with similar features (DPP8/9 cleavage site followed by an IBM), most of them localized to the cytosol. Among them was also the translation initiation factor EIF2A. Its different N-terminal variants are displayed on the right. (C) Strategy to identify N-termini of EIF2A in HEK293 cells. Cells were incubated with the proteasome inhibitor MG132 for 6 h. After lysis, endogenous EIF2A was immunoprecipitated and the resulting precipitate analyzed by mass spectrometry. (D) In intact cells, four different N-terminal peptides of EIF2A were identified by MS analysis: processed acetylated and free NH₂-STPLLTVR, NH₂-APSTPLLTRVR and NH₂-MAPSTPLLTRVR. EIF2A was immunoprecipitated from HEK293 WT and DPP8/9 KO cells treated with MG132 6 h before lysis and precipitates then analyzed by MS. The raw intensities of the four N-terminal peptides comparing HEK293 WT with DPP8/9 KO cells are plotted as bar charts for each peptide. For "peptides w/o N-terminus" mean intensities of non-N-Terminal peptides (WT $n = 54$; KO $n = 55$) of EIF2A were calculated and plotted. Experimental replicates $n = 2$. (E) Peptide constructs used for interaction screen. Various peptides containing amino acid residues representing different EIF2A termini were used for an interaction screen using HeLa cell lysate. (F) EIF2A processed by MAP and DPP8/9 but not unprocessed EIF2A interacts with IAPs. Peptides with the respective N-terminal peptide of EIF2A were incubated with HeLa cell lysates. Interaction partners were identified using MS. Pairwise, two-sided Student's $t$ test was used for statistical testing. $n = 3$ biological replicates. (G) EIF2A processed by MAP and DPP8/9 but not MAP-processed AK2 interacts with IAPs. Peptides with the respective N-terminal peptide of EIF2A were incubated with HeLa cell lysates. Interaction partners were identified using MS. Pairwise, two-sided Student's $t$ test was used for statistical testing. $n = 3$ biological replicates. (H) Acetylation of the N-terminus of EIF2A inhibits the interaction with IAPs. Either acetylated or non-acetylated N-terminal EIF2A peptides were incubated with HeLa cell lysates. Interaction partners were identified using MS. (I) Depletion of NatA levels destabilizes EIF2A2. The levels of the catalytic domain of NatA were depleted using siRNA in HeLa cells. Cell lysates were analyzed via SDS-PAGE and immunoblot. $n = 3$ biological replicates. (J) XIAP ubiquitinates the DPP8/9-processed free N-terminus of EIF2A in vitro. The experiment was performed as explained in Fig. 3H. The indicated N-terminal peptides of EIF2A were subjected to an in vitro ubiquitination assay using purified GST-XIAP. An example of an immunoblot is shown in the left panel. In the right panel, the ratio of $-/+$ XIAP of quantified immunoblots ($n = 2$) for each peptide normalized on streptavidin amounts is plotted. (K) Model. Different strategies exist to mask IBMs in cells. Unmasked IBMs result in rapid degradation of the protein and potentially sensitization of cells toward apoptosis. Equipping the pro-proteins SMAC/DIABLO and HtrA2 with bpMTS for import into the IMS blocks the IBM during import into the IMS. Cleavage of the bpMTS inside mitochondria unmasks the IBM, which is however not accessible to cytosolic IAPs until release of the factors is initiated during apoptosis induction. Likewise, proteins are masked by DPP8/9-cleavable peptides until these peptidases process the proteins. Lastly, N-terminal acetylation of exposed IBMs prevents their binding to IAPs. Source data are available online for this figure.

Collectively, we demonstrated that the DPP8/9-processed free N-terminus of AK2 serves as IBM that can be bound by IAPs resulting in ubiquitination and degradation of AK2. N-terminal acetylation protects AK2 from recognition by IAPs and degradation.

## DPP8/9 processing has the potential to unmask a variety of IAP-binding motifs

DPP8/9 processing unmasks an IBM in AK2. We next searched the human genome (35,589 unique consensus coding sequences (CCDS, NCBI Release 24) for 19,095 genes) for additional proteins with such features (Fig. 4A). This bioinformatic screening revealed that the human genome contains about 2441 proteins (CCDS, 1580 unique genes) with an IBM directly behind the N-terminal methionine corresponding to slightly more than 5% of the dataset. A total of 3465 proteins (1787 (MXP) and 1678 (MP)) contained potential DPP8/9 cleavage sites. Among those, we found that 129 proteins (86 unique genes) contained an IBM behind a DPP8/9 cleavage site (Appendix Fig. S7), indicating that unmasking of an IBM might be a widespread mechanism to regulate the interaction of proteins with IAPs and protein stability (Fig. 4B).

To test whether the concept of IBM unmasking by DPP8/9 is generalizable beyond AK2, we characterized one further candidate identified in the bioinformatic screen. The translation initiation factor EIF2A harbors a DPP8/9 cleavage site followed by an IBM (Fig. 4B). First, we enriched endogenous EIF2A by immunoprecipitation from WT and DPP9 KO cells incubated with MG132 and analyzed the precipitate by mass spectrometry (Fig. 4C). We indeed identified peptides representing the MAP- and DPP8/9-processed N-terminus of EIF2A in the acetylated and non-acetylated forms (NH₂-STPLLTVR; Fig. 4D; Appendix Fig. S8). Lack of DPP9 strongly decreased the amounts of these peptides. We also identified peptides that were DPP8/9 and MAP-unprocessed (Fig. 4D). Amounts of the former were increased in DPP9 KO.

We then tested whether the IBM in EIF2A that became exposed by MAP and DPP8/9 processing ("STPL") was capable of binding IAPs. To this end, we employed the peptide pull-down assay and isolated streptavidin-bound peptides (Fig. 4E) after incubation with HeLa lysate. We then identified interaction partners by mass spectrometry which revealed that NH₂-STPLLTVRGS-KK-Biotin (representing the MAP- and DPP8/9-cleaved N-terminus of EIF2A) efficiently coprecipitated the IAPs XIAP, BIRC2, BIRC3, and BIRC6. Conversely, peptides representing the un- or partially processed EIF2A N-terminus (NH₂-MAPSTPLLTV-KK-Biotin and NH₂-APSTPLLTVR-KK-Biotin) (Fig. 4F,G) did not. Likewise, a peptide representing the acetylated MAP- and DPP8/9-processed EIF2A did not also coprecipitate XIAP, BIRC2, BIRC3, or BIRC6 (Fig. 4H). In line with the absence of IAP binding by acetylated peptides, we demonstrated that depleting NatA by siRNA decreased cellular EIF2A levels (Fig. 4I).

We then assessed the ubiquitination of EIF2A peptides in the in vitro ubiquitination assay, and found that only the NH₂-STPL… peptide representing the free DPP8/9-processed EIF2A N-terminus could be ubiquitinated above background in the presence of XIAP (Fig. 4J, lane 4). Taken together, these assays provide evidence for IAP-mediated degradation of EIF2A following IBM unmasking by DPP8/9 emphasizing that DPP8/9-facilitated unmasking of IBMs appears to be a widespread phenomenon in the regulation of cellular protein stability (Fig. 4K).

## Conclusion

IBMs are important determinants of protein stability and contribute to the regulation of cellular processes, including apoptosis, inflammation, and cell cycle progression. While slightly more than 5% of all proteins carry IBMs directly behind their start methionine, in some proteins, like SMAC/DIABLO and HtrA2, IBMs become exposed only after processing of their bpMTS in the

IMS (Burri et al, 2005; Du et al, 2000; van Loo et al, 2002; Verhagen et al, 2000). Here, we report on the discovery of a class of proteins that contain IBMs initially masked by a DPP8/9-cleavable dipeptide on their N-terminus. For AK2 and EIF2A, unmasking of the IBM by DPP8/9 processing results in the capacity to bind to IAPs. For AK2, this IAP binding triggers rapid degradation in the cytosol. Since DPP8/9 activity differs between tissues and differentiation states and can be regulated by posttranslational modifications on its cysteines or through binding of SUMO, it is tempting to speculate that this class of IBM-containing proteins encounters a further layer of regulation that depends on DPP8/9 activity (Han et al, 2015; Matheeussen et al, 2013; Pilla et al, 2012; Zhang et al, 2013).

A mechanism that prevents IAP binding is the N-terminal acetylation of the IBM by NatA. We found N-terminal acetylation of EIF2A and AK2 at the DPP8/9-processed neo-N-terminus. This is interesting as N-terminal acetylation for many proteins takes place in a co-translational manner (Aksnes et al, 2019), which would suggest that also DPP8/9 processing can take place co-translationally. So far, no association of DPP8/9 with ribosomes or nascent chains was reported and it would be interesting to explore this in the future.

SMAC/DIABLO and HtrA2 expose IBMs after IMS import and subsequent processing. These IBMs become critical upon release of SMAC/DIABLO and HtrA2 during apoptosis. Also, for AK2 a role in apoptosis has been described (Köhler et al, 1999; Lee et al, 2007; Muñoz-Pinedo et al, 2006; Single et al, 1998); however, the proposed mechanism would not require interaction with IAPs. Thus, it remains unclear whether exposure of an IBM after DPP8/9

processing in AK2 contributes to apoptosis or whether it mainly serves in the control of AK2 degradation.

Lastly, we found a subset of 129 proteins that contain DPP8/9 cleavage site-masked IBMs. Among them are besides AK2 and EIF2A, many metabolic enzymes. DPP8/9 have been linked to metabolic rewiring (Finger et al, 2020; Wilson et al, 2013; Zhang et al, 2015), and it will be exciting to test in the future whether these proteins are DPP8/9 targets and whether IBM exposure serves regulatory purposes during differentiation and shifts in metabolism. By comparing our list to the results of experimental DPP8/9-substrate screens (Wilson et al, 2013; Zhang et al, 2015), we further noted that the protein encoded by S100A10 is a validated substrate of DPP9. Furthermore, there is experimental evidence supporting the protein disulfide-isomerase encoded by TXNDC5 and proteins of the semaphorin family (encoded by mouse genes Plxna1 and Sema6c, whereas in humans, we detected PLXNA3 and SEMA4F) as substrates.

Taken together, we provide evidence for a degradation route followed by a subset of DPP8/9 substrates. Interestingly, the N-terminus of mouse AK2 does not significantly bind to IAPs, but instead interacts with UBR following a different degradation pathway (Shimshon et al, 2024). Thus, it appears that different mechanisms have evolved to ensure proper modifications and degradation of subgroups of DPP8/9 substrates, and it will be interesting to uncover common features of different groups of substrates.

# Methods

**Reagents and tools table**

| Reagent/resource | Reference or source | Identifier or catalog number |
|---|---|---|
| **Experimental models** | | |
| Hek293 Flp-In T-Rex (*H. sapiens*) | Thermo Fischer Scientific | R78007 |
| Hek293 Flp-In T-REx-WT-AK2 (*H. sapiens*) | Finger et al, 2020 | N/A |
| Hek293 Flp-In T-REx-WT-AK2-3CS (*H. sapiens*) | Finger et al, 2020 | N/A |
| Hek293 Flp-In T-REx-WT-AK2-S4V,3CS (*H. sapiens*) | Finger et al, 2020 | N/A |
| Hek293 Flp-In T-REx-WT-AK2 P6W (*H. sapiens*) | This study | N/A |
| Hek293 Flp-In T-REx-DPP9 knockout (*H. sapiens*) | Bolgi et al, 2022 | N/A |
| Hek293 Flp-In T-REx-DPP9 knockout-AK2-3CS (*H. sapiens*) | This study | N/A |
| HeLa Flp-In T-REx-AK2 KO (*H. sapiens*) | This study | N/A |
| HeLa Flp-In T-REx-AK2 KO- Mock (*H. sapiens*) | This study | N/A |
| HeLa Flp-In T-REx-AK2 KO- AK2 C40,42,92S (*H. sapiens*) | This study | N/A |
| HeLa Flp-In T-REx-AK2 KO- AK2 S4P,C40,42,92S (*H. sapiens*) | This study | N/A |
| Hek293 Flp-In T-REx-AK2 KO (*H. sapiens*) | This study | N/A |
| Hek293 Flp-In T-REx-293-AK2 KO-AK2 (*H. sapiens*) | This study | N/A |
| HeLa Flp-In T-REx-AK2 KO- SmacMTS AK2 Δ1–3 (*H. sapiens*) | This study | N/A |
| Hek293 Flp-In T-REx-AK2 KO-AK2 C232S (*H. sapiens*) | This study | N/A |
| Hek293 Flp-In T-REx-AK2 KO-AK2 A2C,C232S (*H. sapiens*) | This study | N/A |
| HeLa (*H. sapiens*) | DSMZ (German Collection of Microorganisms and Cell Cultures) | ACC 57 |
| **Recombinant DNA** | | |
| pcDNA5_AK2-HA | Finger et al, 2020 | N/A |
| pcDNA5_AK2-C40,42,92S-HA | Finger et al, 2020 | N/A |

| Reagent/resource | Reference or source | Identifier or catalog number |
|---|---|---|
| pcDNA5_AK2-S4P,C40,42,92S-HA | Finger et al, 2020 | N/A |
| pcDNA5_AK2-C232S-HA | This study | N/A |
| pcDNA5_AK2-A2C,C232S-HA | This study | N/A |
| pcDNA5_AK2-P6W-HA | This study | N/A |
| pcDNA5_SmacMTS-AK2-Δ1–3 | This study | N/A |
| pSpCas9(BB)-2A-GFP (PX458) | Addgene (Ran et al, 2013) | Cat# 48138 |
| **Antibodies** | | |
| Rabbit polyclonal anti-HA | Sigma-Aldrich | Cat# SAB4300603; RRID:AB_10620829 |
| Mouse monoclonal anti-LDH (H-10) | Santa Cruz | Cat# sc-133123; RRID:AB_2134964 |
| Rabbit polyclonal anti-AK2 | Finger Habich et al, 2020 | N/A |
| Rabbit polyclonal antiCPOX | St John's Laboratory | Cat# STJ23214 |
| Rabbit polyclonal antiDPP9 | abcam | Cat# ab42080 |
| Rabbit polyclonal antiPSMC4 | Sigma-Aldrich/Merck | Cat# HPA002044-100UL |
| Mouse monoclonal anti-actin | ThermoFisher Scientific | Cat# #MA5-11869 |
| Rabbit polyclonal antiXIAP | Sigma-Aldrich/Merck | Cat# SAB5700239 |
| Rabbit polyclonal antiStreptavidin | Rockland antibodies & assays | Cat#100-4195 |
| Rabbit polyclonal antiArd1A (NatA) | Invitrogen | Cat# PA5-32236 |
| Rabbit polyclonal anti-EIF2A/CDA02 | Proteintech | Cat# 11233-1-AP |
| Mouse monoclonal anti-ubiquitin-HRP | Santa Cruz Biotechnology | Cat# sc-53509 |
| Rat monoclonal antiBIRC2 | Enzo | Cat# ALX-803-335 |
| Mouse monoclonal antiVinculin | Sigma-Aldrich | Cat#V9131 |
| Goat anti-Mouse IgG (H&L), HRP Conjugate | ImmunoReagents | Cat# GtxMu-003-DHRPX |
| Goat anti-Rabbit IgG (H&L), HRP Conjugate | ImmunoReagents | Cat# GtxRb-003-DHRPX |
| **Oligonucleotides and other sequence-based reagents** | | |
| AK2 C232S-HA primer fw (5′ -3′) CCTTCTCCAAAGCCACATCTAAAGACTTGGTTATG | This study | N/A |
| AK2 C232S-HA primer rv (5′-3′) CATAACCAAGTCTTTAGATGTGGCTTTGGAGAAGG | This study | N/A |
| AK2 A2C,C232S-HA primer fw (5′-3′) CAGGGTACCGCAGCCatgTGTCCCAGCGTG | This study | N/A |
| AK2 A2C,C232S-HA primer rv (5′-3′) ACTTGGTTATGTTTATCtatccttatgatgtacctgattatgcataaGCGGCCGCAGC | This study | N/A |
| AK2 P6W-HA primer fw (5′-3′) CAGGGTACCGCAGCCatgGCTCCCAGCGTGTGGGCGGCAGAACCCGAG | This study | N/A |
| AK2 P6W-HA primer rv (5′-3′) ACTTGGTTATGTTTATCTATCCTTATGATGTACCTGATTATGCATAAGCGGCCGCAGC | This study | N/A |
| SmacMTS AK2 Δ1–3 primer fw (5′-3′) CAGCTTAAGGCAGCCATGGCGGCTCTG | This study | N/A |
| SmacMTS AK2 Δ1–3 primer fw (5′-3′) ACTTGGTTATGTTTATCTATCCTTATGATGTACCTGATTATGCATAAGCGGCCGCAGC | This study | N/A |
| AK2 Guide #3 sgRNA2 Fw: CACCGCTACCATTTCATCACTCACC | This study | N/A |
| AK2 Guide #3 sgRNA2 Rev: AAACGGTGAGTGATGAAATGGTAGC | This study | N/A |
| XIAP siRNA AAGTGCTTTCACTGTGGAGGA | Qiagen | Cat# SI00299446 |
| XIAP siRNA CGAGCAGGGUUUCUUUAUAtt | Sigma-Aldrich | HA14860896 |
| XIAP siRNA UAUAAAGAAACCCUGCUCGtg | Sigma-Aldrich | HA14860897 |
| NatA siRNA AACTTTCAGATCAGTGAAGTG | Qiagen | Cat# SI00299243 |

| Reagent/resource | Reference or source | Identifier or catalog number |
|---|---|---|
| NatA siRNA ATCAGTGAAGTGGAGCCCAAA | Qiagen | Cat# SI03047100 |
| NatA siRNA GUUAUGCAAUGAGUACUGAtt | Sigma-Aldrich | HA14860898 |
| NatA siRNA UCAGUACUCAUUGCAUAACtg | Sigma-Aldrich | HA14860899 |
| **Chemicals, enzymes, and other reagents** | | |
| 1G244 | Sigma-Aldrich | SML2247-5MG |
| MG132 | Sigma-Aldrich | C2211-5MG |
| Emetine dihydrochloride hydrate | Sigma-Aldrich | Cat # 7083-71-8 |
| One Shot TOP10 Chemically Competent *E. coli* | Thermo Fisher | Cat# C404010 |
| **Software** | | |
| Alphafold Multimer | Jumper et al, 2021 | N/A |
| ChimeraX (version 1.4) | https://www.cgl.ucsf.edu/chimerax/ | N/A |
| Pymol | https://www.pymol.org | N/A |
| MaxQuant | https://www.maxquant.org/ | N/A |
| DIA-NN | https://github.com/vdemichev/DiaNN/releases/tag/ | N/A |
| **Other** | | |
| N/A | N/A | N/A |
| N/A | N/A | N/A |

## Plasmids, cell lines, and chemical treatments of cells

For plasmids, cell lines, chemicals, antibodies and other tools used in this study, see Reagents and Tools Table. Cells were cultured in DMEM supplemented with 10% fetal calf serum (FCS) and penicillin/streptomycin (Pen/Strep) at 37 °C under 5% $CO_2$. For Emetine chase experiments, cells were treated for indicated times with 100 µg/ml emetine (dissolved in water). The cells were treated with 1 ml DMEM with 100 µg/ml emetine. After the indicated time, the cells were washed with 0.5 ml ice-cold PBS and harvested in reducing Laemmli buffer. For the generation of stable, inducible cell lines the HEK293 cell line–based Flp-In T-REx-293 cell line was used with the Flp-In T-REx system (Invitrogen). For MG132 treatment, the cells were treated with 5 µM MG132 in 1 ml DMEM for 6 h.

## Generation of stable inducible cell lines

For the generation of stable cell lines with different AK2 variants, the inducible Flp-In T-REx System was used. All AK2 constructs were cloned into the pcDNA5 FRT-TO vector and co-transfected with the pOG44 Vector into the different host cell lines by using the transfection reagent FuGene, according to the manufacturer's guideline. Positive clones were selected with glucose-containing medium (DMEM supplemented with 10% FCS and 500 mg/ml Pen/Strep) containing 10 µg/ml Blasticidin and 100 µg/ml Hygromycin. Expression of constructs was induced using 1 µg/ml doxycycline for 16 h.

## Western blot image acquisition

The immunoblotting images were detected using the ChemiDoc Touch Imaging system (Bio-Rad).

## Native immunoprecipitation

To detect protein-protein interactions, native immunoprecipitation (IP) was performed. Corresponding cell lines were seeded on a 15-cm dish and were grown until a confluency of 90%. Before IP was performed, inducible cell lines were induced with doxycycline overnight. Then the cells were washed with 5 ml ice-cold PBS. The cells were scratched off and pelleted at $500 \times g$ for 3 min at 4 °C. The pellet was resuspended in 1 ml native IP buffer (25 mM NaCl, 100 mM NaPi pH 8.1; 0% Triton X-100) and was sonified for lysis. After that, the samples were centrifuged at $20,817 \times g$ for 1 h at 4 °C. 750 µl of the supernatant was collected and incubated with 10 µl equilibrated HA beads (monoclonal anti-HA-Agarose produced in mouse) for 1 h at 4 °C. Also, 75 µl were taken separately as a total sample. The beads were washed 6 times with 1 ml IP buffer with a centrifugation step at $2000 \times g$ for 1 min at 4 °C. After the last centrifugation step, the beads were dried completely and 30 µl reducing Laemmli was added and the samples were boiled twice for 10 min. The samples were analyzed by SDS-PAGE and western blot.

## Fractionation of HEK293 cells

For the fractionation of HEK293 cells, cells were cultivated for 2 days on 15-cm dishes. For harvesting, the cells were washed 1 time with 10 ml ice-cold 1 × PBS and scraped off using a cell scraper. Afterward, the cells were resuspended in mito buffer (20 mM HEPES pH 7.4, 220 mM Mannitol, 70 mM Sucrose, 1 mM EDTA) and homogenized using a 2 ml homogenizer and then centrifuged at $800 \times g$ for 5 min at 4 °C. In total, 200 µl of the supernatant was taken as total, proteins were precipitated using TCA. The residual supernatant was transferred

into a new reaction tube and centrifuged at 13,000 × g for 15 min. The pellet was washed several times with mito buffer and then resuspended in mito buffer, proteins were precipitated via TCA. The supernatant was centrifuged at full speed for 10 min. Proteins of the resulting supernatant were precipitated using TCA. TCA precipitations were frozen for 2 h at −20 °C.

After thawing the precipitate was washed and dissolved in Buffer A (6 M urea, 0.2 M Tris pH 7.5, 10 mM EDTA, 2% SDS).

## Assay to detect processing events via protein thiols (maleimide shift assay)

To visualize the processing events of AK2 in different cell lines, 300,000 cells were seeded on a six-well dish. Before the cells were harvested, they were treated for 16 h with doxycycline (1 μg/ml) to induce AK2 construct expression. For harvesting, the cells were washed with 1 ml ice-cold PBS and the non-reducing Laemmli buffer (2% SDS, 60 mM Tris, pH 6.8, 10% glycerol, 0.0025% bromphenol blue) containing 10 mM TCEP. After incubation on ice for 50 min, the samples were sonified. In all, 250 mM of mmPEG12 was added. After incubation for 1 h on ice in the dark, samples were loaded and separated on SDS-PAGE, analyzed by western blotting and immunodetection.

## siRNA-mediated knockdown

Reverse siRNA transfections were performed in 12-well plates using RNAiMAX following the instructions of the manufacturer. In short, 27.5 pmol siRNA and 1.47 μl Lipofectamine™ RNAiMAX were mixed with 100 μl Opti-MEM® (FisherScientific) and incubated for 10 min. After pipetting the transfection mix into the wells, 60k of HeLa cells in 1 ml DMEM (supplemented with 10% FCS) were added into the well (final siRNA concentration of 25 nM). Cells were incubated at 37 °C for 72 h. In case cell lines overexpressing certain proteins were used, overexpression was induced by adding 1 ml induction media for an additional 24 h. Protein levels were analyzed by immunoblotting.

## Bioinformatic screen

To assess IBM motifs across the human genome, we retrieved the current set of consensus coding sequences (CCDS) from the NCBI CCDS project (Release 24, October 2022). N-terminal in silico processing was performed with the following rules: MAP cleavage: M|[^D]; DPP9 "long": M[^D^E]P|[^P]; DPP9 "short": MP|[^P]; with characters following a caret sign ("^") being excluded in the respective position and cleavage at the "|" sign. Resulting neo-N-termini were assessed for the presence of an IBM motif by the following pattern: [AS][DEFGILMQRSTV][ACGKMPRSV][ADEF-GILVWY], allowing any of the characters in parentheses at the respective position of the N-terminal 4-amino acid sequence. CCDS were mapped to Ensembl gene names using Ensembl BioMart and were checked for duplicates, indicating multiple CCDS corresponding to the same gene. We then compared our results to candidate DPP8/9 substrates derived from studies using experimental methods. Specifically, we focused on 29 candidate DPP8/DPP9 substrates identified by parsing TAILS N-terminomics data (Wilson et al, 2013) and 111 candidate proteins detected by subjecting DPP9-enzyme-inactive MEF cells to in vitro DPP9 digestion and analysis by 2D-DIGE-mass spectrometry (Zhang et al, 2015).

## Quantification and statistical analysis

The intensity of autoradiography and immunoblot signals were quantified using Image Lab (Biorad). Error bars in Figures represent standard deviation. The number of experiments is reported in the Figure legend.

## EIF2A IPs to asses ubiquitination

Reverse siRNA transfections were performed using Lipofectamine™ RNAiMAX as described above. In six-well plates, HEK293 WT and DPP9 KO cells were treated with 25 nM siRNA oligos for XIAP and BIRC2 or without oligos for 48 h. In addition, MG132 or corresponding amounts of DMSO were added for 6 h before lysis after 48 h to WT and KO cells treated with and without RNAi. Cells were harvested and lysed in lysis buffer (150 mM KCl, 75 mM HEPES, pH 7.5, 1.5 mM EGTA, 1.5 mM MgCl$_2$, 10% glycerol, 1 mM DTT and 0.1% NP-40) supplemented with protease inhibitor cocktail (Serva, Heidelberg, Germany) and PhosSTOP phosphatase inhibitors (Roche, Basel, Switzerland) and diluted to 1 μg/μl. Anti-EIF2A antibody (1 μl) (# 11233-1-AP, Proteintech, IL, USA) was bound to Protein A beads (40 μl slurry) using the Dynabeads™ Protein A Immunoprecipitation Kit (# 10006D, Invitrogen, Waltham, MA, USA) in 200 μl of Ab Binding & Washing Buffer for 1 h at room temperature. Then, 200 μg of cellular lysates (200 μl) were incubated with the antibody-conjugated beads rotating overnight at 4 °C. Next day, the beads were washed five times: twice with low salt buffer (50 mM HEPES pH 7.4, 140 mM NaCl, 1% Triton), twice with high salt buffer (50 mM HEPES pH 7.4, 500 mM NaCl, 1% Triton) and once with TBS (50 mM Tris pH 7.4, 150 mM NaCl). IPs were blotted against ubiquitin and EIF2A. The input material was blotted against EIF2A, XIAP, BIRC2, and Vinculin.

## Peptide pull-downs

Peptide pull-downs were performed as previously described (Mueller et al, 2021; Muller and Bange, 2023). Briefly, streptavidin beads were incubated with biotinylated peptides (10 μg peptide/ 20 μl slurry beads) for 30 min at RT in binding buffer A (150 mM NaCl, 50 mM Tris pH 8.0, 0,075% NP-40). Beads were washed once with lysis buffer (150 mM KCl, 75 mM HEPES, pH 7.5, 1.5 mM EGTA, 1.5 mM MgCl$_2$, 10% glycerol, 1 mM DTT, and 0.1% NP-40) supplemented with protease inhibitor cocktail (Serva, Heidelberg, Germany) and PhosSTOP phosphatase inhibitors (Roche, Basel, Switzerland). After washing, 1 mg of HeLa lysate in lysis buffer (2 mg/ml) was added to each sample. Pull-downs were washed twice with lysis buffer and twice with wash buffer (lysis buffer without NP-40, glycerol, DTT, inhibitor cocktail and PhosSTOP phosphatase inhibitors). All experiments were performed in triplicates. Afterward, samples were transferred to new tubes and processed for mass spectrometry analysis.

## Preparation of the whole proteome of HEK293 WT and DPP9 KO cells for MS

HEK293 WT and DPP9 KO cells were grown as mentioned above. Cells were lysed in 2% SDC and digested with Trypsine (1:100 enzyme: protein ratio) and LysC (1:100 enzyme: protein ratio)

overnight. 100 µg of digested HEK293 WT and DPP9 KO lysates were fractionated with the high-pH fractionation kit from Pierce (#84868, ThermoFisher Scientific) and directly measured by MS.

## IPs for N-term identification of EIF2A

For the identification of EIF2A N-terminal peptides, HEK293 WT and DPP9 KO cells were grown until a confluency of 90% and treated with the proteasome inhibitor MG132 for 6 h before lysis. Cells were harvested in lysis buffer (150 mM KCl, 75 mM HEPES, pH 7.5, 1.5 mM EGTA, 1.5 mM MgCl$_2$, 10% glycerol, 1 mM DTT and 0.1% NP-40) supplemented with protease inhibitor cocktail (Serva, Heidelberg, Germany) and PhosSTOP phosphatase inhibitors (Roche, Basel, Switzerland) and diluted to 2 µg/µl. Anti-EIF2A antibody (4 µl) (# 11233-1-AP, Proteintech, IL, USA) was bound to Dynabeads$^{Tm}$ Protein A Kit beads (40 ul slurry) (# 10006D, Invitrogen, Waltham, MA, USA) in 200 µl of lysis buffer for 1 h at room temperature. Then, 1 mg of cellular lysates were incubated with the antibody-conjugated beads rotating overnight at 4 C. The beads were then washed five times, twice with low salt buffer (50 mM HEPES pH 7.4, 140 mM NaCl, 1% Triton), twice with high salt buffer (50 mM HEPES pH 7.4, 500 mM NaCl, 1% Triton) and 1× with TBS (50 mM Tris pH 7.4, 150 mM NaCl). In all, 1/10 of the beads was kept for a western blot, and the rest was prepared for MS analysis.

## In vitro ubiquitination

GST-XIAP was purified and in vitro ubiquitination assays were performed as previously described (Mueller et al, 2021). Briefly, the peptides also used in the peptide pull-down assay (Fig. 2D) were bound to Streptavidin beads, washed and subjected to in vitro ubiquitination assays. For each assay, 500 ng of peptides were used. Final concentrations for the in vitro ubiquitination assay were 50 nM E1, 250 nM E2 (UBCH5A), 1 µM GST-XIAP, 3 ug ubiquitin, 3 mM Mg-ATP and ligase buffer (apart from GST-XIAP, reagents were purchased from Boston Biochem/ R&D Systems; Minneapolis, USA). In vitro ubiquitination reactions were incubated for 1 h at 30 °C and then immediately washed five times with 500 µl wash buffer (150 mM KCl, 75 mM HEPES, pH 7.5, 1.5 mM EGTA, 1.5 mM MgCl$_2$, 10% glycerol, 1 mM DTT and 0.1% NP-40) containing additionally 1% Triton to remove XIAP from the beads. Ubiquitination of peptides was assessed by immunoblotting anti-ubiquitin.

## Mass spectrometric analysis

Peptide pull-downs and IPs were reduced, alkylated and digested directly on beads as previously described (Mueller et al, 2021; Muller and Bange, 2023). Obtained peptides of AK2, COMMD10 and STXBP2 were separated with a PepMap100 RSLC C18 nano-HPLC column (2 µm, 100 Å, 75 ID x 25 cm, nanoViper, Dionex, Germany) on an UltiMate$^{TM}$ 3000 RSLCnano system (Thermo-Fisher Scientific, Waltham, Massachusetts, United States) using a 125 min gradient from 5–60% acetonitrile with 0.1% formic acid and then directly sprayed via a nano-electrospray source (Nanospray Flex Ion Source, Thermo Scientific, Waltham, MA, USA) in a Q Exactive$^{TM}$ Plus Hybrid Quadrupole-Orbitrap Mass Spectrometer (ThermoFisher Scientific, Waltham, MA, USA). The Q Exactive$^{TM}$

Plus was operated in a data-dependent mode acquiring one survey scan and subsequently ten MS/MS scans. A mass range of m/z 300–1650 was acquired with a resolution of 70,000 for full scan, followed by up to ten high energy collision dissociation (HCD) MS/MS scans of the most intense at least doubly charged ions. Obtained peptides of EIF2A peptide pull-downs, IPs and AK2 IPs were separated on an EASY-nLC 1200 HPLC system (ThermoFisher Scientific) using an 80 min gradient from 5–60% acetonitrile with 0.1% formic acid. Peptides were directly sprayed via a nano-electrospray source in an Q Exactive HF-X Spectrometer (Thermo-Fisher Scientific). Data were acquired in a data-independent mode acquiring one survey scan (MS scan) and then 46 windows of 20.1 m/z isolation width for MS2 scans. A mass range of m/z 300–1000 was acquired with a resolution of 120.000 for full scans (AGC target 3 × E6). MS2 scans were recorded with a resolution of 30,000 (AGC target 1 × E6). Fractions of whole cellular lysates from HEK293 WT and DPP8/9 KO cells were measured with the same settings.

## MS data analysis

The resulting raw files of peptide pull-downs of AK2, COMMD10, and STXBP2 were processed with the MaxQuant software (version 1.6.14) searching against a UNIPROT human database (ref. Jan 2021) using oxidation (M) and acetylation (N-terminus) as variable modifications and carbamidomethylation (C) as fixed modification (Cox and Mann, 2008). A false discovery rate cut-off of 1% was applied at the peptide and protein level. The integrated label-free algorithm was used for the relative quantification of proteins (Cox et al, 2014). Peptide pull-downs of EIF2A and immunoprecipitations of EIF2A were analyzed with DIA-NN (version 1.9.1) (Demichev et al, 2020) using a UNIPROT human database (ref. Jan 2023) as a library. Ox (M), N-term M excision, Ac(N-term), C carbamidomethylation were used as variable modifications allowing up to three variable modifications and one miscleavage. Precursor FDR was set to 1%. Immunoprecipitations of EIF2A were searched once with standard settings Trypsin/P and another time using the command line "—cut P*, K*, R*, !*P" under additional options to allow cleavage after proline and identification of the DPP8/9-cleaved N-terminus of EIF2A. In addition, the raw files were searched against a modified human database where proteins starting with MP or MXP (X = any amino acid) were modified by removing MP or MPX, while others remained in full-length to allow identification of acetylated DPP89 processed. Quantified N-terminal and internal peptides of EIF2A IPs have been extracted from the report.pr_matrix DIA-NN output and can be found as extended data sets. Whole proteome data HEK293 WT and DPP8/9 KO cells were with DIA-NN (version 1.9.1) using the same setting and same searches as for EIF2A IPs (human FASTA, additional cut after P, modified human database removing MP and MXP at the N-terminus of proteins from the file. Quantified N-terminal and internal peptides of AK2 have been extracted from the report.pr_-matrix DIA-NN output and can be found as extended datasets.

Peptide spectral data from the EIF2A IPs generated from DIA-NN (version 1.9.1) were imported into the Skyline-daily (version 24.0.9.171, MacCoss Lab, University of Washington, USA) under the proteomics interface with the following settings: (1) q value threshold set at 0.05; (2) workflow set to the DIA; (3) iRT standard peptides set to "CiRT(iRT-C18)". All the filtered peptides were re-

calibrated by Skyline-daily and searched in the chromatogram. Posttranslational modifications (PTMs) setting included: (1) Oxidation (M); (2) Acetyl (N-term); (3) Carbamidomethyl (C). The scanning of peptide ion searching setting were: (1) Precursor ion charges: 1, 2, 3; (2) Ion charges: 1, 2; (3) Ion types: y,b,p; (4) *m/z* range: 50–2000; (5) Mass accuracy of MS1 filtering and MS/MS filtering: 20 ppm; (6) Retention time filtering: "Within 5 min of predicted RTs". The filtered peptide ions were then searched against a human proteome FASTA file (UniProt, accessed May, 2023) or a DPP-modified FASTA file (MP/MXP cleaved on the N-termini), with the cleavable sites set to K/R and a maximal of 1 missing cleavage. The MS/MS spectra of peptides of interest (EIF2A N-termini from IPs and AK2 N-termini from whole cellular lysates) were generated at their peak retention times.

## Bioinformatic data analysis of peptide pull-downs

Quantified proteins from peptide pull-downs were further analyzed with Perseus (version 1.6.50) (Tyanova et al, 2016). Contaminant and reverse hits were removed and LFQ intensities logarithmized. All experiments were at least done in triplicates and proteins were required to be quantified in at least two out of three replicates to be considered. Afterward, missing values were imputed at the lower end of the distribution of all measured and quantified proteins (downshift 1.8, width 0.3). Specific interactors were identified with a Student's two sample *t* test with a cut-off of 2 for the decadal logarithm of the *P* value. For IPs anti-HA AK2 we required quantification in three out of three replicates to be considered. Data were imputed at the lower end of the distribution (downshift 2, width 0.3).

## In silico structural analysis of AK2 and XIAP

To obtain a model of the processed N-terminus of AK2 interacting with the BIR3 domain of XIAP, residues 4–10 of human AK2 and 265–330 of human XIAP were used as an input for Alphafold 3 (Abramson et al, 2024). The model resulting from the full-length AK2 prediction using Alphafold 3 was colored according to the pLDDT values using the structure prediction tool in ChimeraX (Meng et al, 2023). All models shown have been visualized using ChimeraX. The disorder content of human AK2 was predicted using PrDOS (Ishida and Kinoshita, 2007).

## Data availability

The mass spectrometry proteomics data have been deposited to the ProteomeXchange Consortium via the PRIDE (Perez-Riverol et al, 2022) partner repository with the dataset identifier PXD047178.

The source data of this paper are collected in the following database record: biostudies:S-SCDT-10_1038-S44319-025-00455-z.

## Peer review information

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

## Acknowledgements

The Deutsche Forschungsgemeinschaft (DFG, German Research Foundation) funds research in the Laboratory of JR through the grants RI2150/5-1 project number 435235019, RI2150/2-2 project number 251546152, RTG2550/1 project number 411422114, CRC1678 - project number 520471345, SPP2453 project number 541742459, Major Research Instrumentation project number - 533907460, and CRC1218 - project number 269925409. TB gratefully acknowledges funding by the DFG (project number: 5041 140321) and by the Friedrich-Baur Stiftung. TB is also thankful for funding by the LMU Munich's Institutional Strategy LMUexcellent within the framework of the German Excellence Initiative and MS instrumentation by the DFG (INST 86/1800-1 FUGG for MSR). SP is funded by CMMC core funding (JRG XI), by the DFG-SFB1430-Project-ID 424228829, and the CANTAR network funded by the Ministry of Culture and Science of the state of Northrhine-Westphalia. KJL, KW, and JR thank Kathrin Ulrich and members of the Riemer lab for the critical reading of the manuscript. We thank Anja Wittmann and Anika Seiler for technical support throughout the project. FM, YX, and TB thank A Musacchio, the Bange and Robles group for their support.

## Author contributions

**Kim J Lapacz**: Resources; Investigation; Visualization; Methodology; Writing—review and editing. **Konstantin Weiss**: Conceptualization; Resources; Formal analysis; Investigation; Visualization; Methodology; Writing—review and editing. **Franziska Mueller**: Resources; Investigation; Methodology; Writing—review and editing. **Yuxing Xue**: Formal analysis; Visualization; Methodology. **Simon Poepsel**: Investigation; Visualization; Methodology; Writing—review and editing. **Matthias Weith**: Data curation; Investigation; Writing—review and editing. **Tanja Bange**: Resources; Formal analysis; Supervision; Funding acquisition; Investigation; Visualization; Methodology; Writing—original draft; Writing—review and editing. **Jan Riemer**: Conceptualization; Resources; Formal analysis; Supervision; Visualization; Writing—original draft; Project administration; Writing—review and editing.

Source data underlying figure panels in this paper may have individual authorship assigned. Where available, figure panel/source data authorship is listed in the following database record: biostudies:S-SCDT-10_1038-S44319-025-00455-z.

## Funding

## Disclosure and competing interests statement

The authors declare no competing interests.

