## [Peer Review File · EMBO Reports]

DPP8/9 processing of human AK2 unmasks an IAP binding motif

Kim Lapacz, Konstantin Weiss, Franziska Mueller, Yuxing Xue, Simon Poepsel, Matthias Weith, Tanja Bange, and Jan Riemer

Corresponding author(s): Jan Riemer (jan.riemer@uni-koeln.de), Tanja Bange (tanja.bange@med.uni-muenchen.de)

Review Timeline:

Submission Date:	19th Dec 23
Editorial Decision:	3rd Jan 24
Appeal Received:	13th Jul 24
Editorial Decision:	13th Aug 24
Revision Received:	31st Dec 24
Editorial Decision:	27th Feb 25
Revision Received:	13th Mar 25
Accepted:	27th Mar 25

Transaction Report:

3rd Jan 2024

Dear Prof. Riemer,

Thank you for the submission of your manuscript to EMBO Reports. I apologize for my delayed response, which is due to the Christmas and New Year holiday season. I have now read and discussed your work with my colleagues here, and I regret to say that we all agree that it is not well suited for our journal.

We appreciate that your study reports that DPP8/9 processing unmasks a hidden IBM in AK2. You find that IAP protein binding destabilizes cytosolic AK2. You further provide evidence that a similar arrangement of an IBM C-terminal to a DPP8/9 cleavage site is found in 129 human proteins.

We acknowledge that your findings indicate that AK2 stability depends on an N-terminal IBM motif, which will be of interest to researchers working in the field. However, we also note that the evidence is currently limited to one protein, for which DPP8/9 processing and subsequent degradation in the cytosol has already been reported. We also note that the concept of unmasking IBMs upon proteolytic processing as such is known. Based on these considerations we think that the manuscript, as it stands, is not sufficiently developed for consideration for publication by EMBO reports. Further evidence that the concerted action of DPP8/9 processing and IBM unmasking is indeed a more general phenomenon, as proposed, would be required for consideration at EMBO Reports, e.g., by verifying a subset of the identified candidate proteins. As the manuscript stands, we have therefore decided not to proceed with in-depth peer review.

That said, your work is an excellent candidate for our partner journal Life Science Alliance (<http://www.life-science-alliance.org/>; our broad scope Open Access journal published in partnership between the EMBO-, Rockefeller University-, and Cold Spring Harbor Laboratory Presses). The editors of Life Science Alliance would be pleased to send your manuscript for in-depth peer review; no reformatting is required.

We very much hope you will be interested in this option: please follow the link below for transfer. Eric Sawey, Executive Editor of Life Science Alliance (e.sawey@life-science-alliance.org), will be pleased to answer any questions.

Yours sincerely,

** As a service to authors, EMBO Press provides authors with the ability to transfer a manuscript that one journal cannot offer to publish to another journal, without the author having to upload the manuscript data again. To transfer your manuscript to another EMBO Press journal using this service, please click on Link Not Available

Universität zu Köln

Universität zu Köln • Biochemie • Zùlpicher Str. 47a • 50674 Köln

To Dr. Martina Rembold
Editor of *EMBO Reports*

Köln, 15.07.2024

Dear Dr. Rembold,

Please find enclosed our manuscript entitled “**DPP8/9 processing of human AK2 unmasks an IAP binding motif**” to be considered for publication in *EMBO Reports*. We initially submitted the manuscript (EMBOR-2023-58683V1) last year and received an editorial rejection letter in January 2024. Since then and upon communication with you, we carefully addressed your concerns and now provide evidence that the mechanism of unmasking IAP-binding motifs by DPP8 and 9 is a general phenomenon that occurs inside intact cells.

Specifically, in the manuscript we identify the mechanism by which the essential enzyme adenylate kinase 2 (AK2), which is processed directly after synthesis at its N-terminus by methionine aminopeptidase (MAP) and subsequently also by the dipeptidyl peptidases DPP8 and 9 (DPP8/9; Finger, Habich et al, *EMBO J* 2020), is degraded. We found that DPP8/9-processing of AK2 unmasks a so-called IAP (inhibitors of apoptosis)-binding motif (IBM), a four peptide motif at the very N-terminus of AK2. This motif allows IAPs, a class of E3 ubiquitin ligases, to bind to AK2 and direct it to degradation. N-terminal acetylation of this motif mediated by the N-acetyltransferase NatA prevents this interaction thereby stabilizing AK2. In a genome-wide *in silico* screen, we identified an additional 129 substrates, in which an IBM becomes potentially unmasked by DPP8/9 processing, and verified the processing, IBM unmasking and IAP-dependent ubiquitylation for a further candidate, EIF2A. Since DPP8/9 activity differs between tissues and differentiation states and can be regulated by posttranslational modifications on its cysteines or through binding of SUMO, it may be that this class of IBM-containing proteins encounters a further layer of regulation that depends on DPP8/9 activity.

We think that our findings will be of high interest to researchers working in the fields of cellular quality control and proteostasis, and they will appeal to the broad readership of *EMBO Reports*.

Mathematisch-
Naturwissenschaftliche
Fakultät

Department für Chemie
Institut für Biochemie

Prof. Dr. Jan Riemer

Zùlpicher Str. 47a
50674 Köln
Deutschland

Telefon +49 221 470-7306

jan.riemer@uni-koeln.de
www.mathnat.uni-koeln.de

We thank you for your consideration of our manuscript and look forward hearing from you.

With kind regards,

Jan Riemer

Dear Jan,

Thank you for the submission of your research manuscript to our journal. We have now received the full set of referee reports that is copied below.

As you will see, the referees acknowledge that the findings are interesting and that the conclusions are overall supported by the data presented but they also raise a number of partially overlapping concerns, such as N-terminal acetylation being a co-translational event, which would contrast your data on processing. I feel that all referee concerns are pertinent and should be addressed. Please supply the mass spectrometry data as spreadsheets, as suggested by referee 1, and deposit these data in a public repository (see point 7 below).

Given the constructive comments, we would like to invite you to revise your manuscript with the understanding that the referee concerns (as detailed above and in their reports) must be fully addressed and their suggestions taken on board. Please address all referee concerns in a complete point-by-point response. Acceptance of the manuscript will depend on a positive outcome of a second round of review. It is EMBO Reports policy to allow a single round of revision only and acceptance or rejection of the manuscript will therefore depend on the completeness of your responses included in the next, final version of the manuscript.

We realize that it is difficult to revise to a specific deadline. In the interest of protecting the conceptual advance provided by the work, we recommend a revision within 3 months (November 13). Please discuss the revision progress ahead of this time with the editor if you require more time to complete the revisions.

I am also happy to discuss the revision further via e-mail or a video call, if you wish.

*****IMPORTANT NOTE:

We perform an initial quality control of all revised manuscripts before re-review. Your manuscript will FAIL this control and the handling will be delayed IN CASE the following APPLIES:

- 1) A data availability section providing access to data deposited in public databases is missing. If you have not deposited any data, please add a sentence to the data availability section that explains that.
- 2) Your manuscript contains statistics and error bars based on $n=2$. Please use scatter blots in these cases. No statistics should be calculated if $n=2$.

When submitting your revised manuscript, please carefully review the instructions that follow below. Failure to include requested items will delay the evaluation of your revision.*****

- 1) a .docx formatted version of the manuscript text (including legends for main figures, EV figures and tables). Please make sure that the changes are highlighted to be clearly visible.
- 2) individual production quality figure files as .eps, .tif, .jpg (one file per figure). Please download our Figure Preparation Guidelines (figure preparation pdf) from our Author Guidelines pages <https://www.embopress.org/page/journal/14693178/authorguide> for more info on how to prepare your figures.
- 3) a .docx formatted letter INCLUDING the reviewers' reports and your detailed point-by-point responses to their comments. As part of the EMBO Press transparent editorial process, the point-by-point response is part of the Review Process File (RPF), which will be published alongside your paper.
- 4) a complete author checklist, which you can download from our author guidelines (<<https://www.embopress.org/page/journal/14693178/authorguide>>). Please insert information in the checklist that is also reflected in the manuscript. The completed author checklist will also be part of the RPF.
- 5) Please note that all corresponding authors are required to supply an ORCID ID for their name upon submission of a revised

manuscript (<<https://orcid.org/>>). Please find instructions on how to link your ORCID ID to your account in our manuscript tracking system in our Author guidelines (<<https://www.embopress.org/page/journal/14693178/authorguide#authorshipguidelines>>)

6) We replaced Supplementary Information with Expanded View (EV) Figures and Tables that are collapsible/expandable online. A maximum of 5 EV Figures can be typeset. EV Figures should be cited as 'Figure EV1, Figure EV2' etc... in the text and their respective legends should be included in the main text after the legends of regular figures.

7) Before submitting your revision, primary datasets (and computer code, where appropriate) produced in this study need to be deposited in an appropriate public database (see <<https://www.embopress.org/page/journal/14693178/authorguide#dataavailability>>).

Specifically, we would kindly ask you to provide public access to mass spectrometry dataset.

The accession numbers and database should be listed in a formal "Data Availability " section (placed after Materials & Method) that follows the model below (see also <<https://www.embopress.org/page/journal/14693178/authorguide#dataavailability>>). Please note that the Data Availability Section is restricted to new primary data that are part of this study.

Data availability

7) Please note that a Data Availability section at the end of Materials and Methods is now mandatory. In case you have no data that requires deposition in a public database, please state so instead of refereeing to the database. See also <<https://www.embopress.org/page/journal/14693178/authorguide#dataavailability>>. Please note that the Data Availability Section is restricted to new primary data that are part of this study.

Additional information on source data and instruction on how to label the files are available <<https://www.embopress.org/page/journal/14693178/authorguide#sourcedata>>.

10) Figure legends and data quantification:
The following points must be specified in each figure legend:

- the name of the statistical test used to generate error bars and P values,
 - the number (n) of independent experiments (please specify technical or biological replicates) underlying each data point,
 - the nature of the bars and error bars (s.d., s.e.m.)
- If the data are obtained from n {less than or equal to} 5, show the individual data points in addition to the SD or SEM.
- If the data are obtained from n {less than or equal to} 2, use scatter blots showing the individual data points.

11) Our journal encourages inclusion of *data citations in the reference list* to directly cite datasets that were re-used and obtained from public databases. Data citations in the article text are distinct from normal bibliographical citations and should directly link to the database records from which the data can be accessed. In the main text, data citations are formatted as follows: "Data ref: Smith et al, 2001" or "Data ref: NCBI Sequence Read Archive PRJNA342805, 2017". In the Reference list, data citations must be labeled with "[DATASET]". A data reference must provide the database name, accession number/identifiers and a resolvable link to the landing page from which the data can be accessed at the end of the reference. Further instructions are available at <<https://www.embopress.org/page/journal/14693178/authorguide#referencesformat>>.

12) All Materials and Methods need to be described in the main text using our 'Structured Methods' format, which is required for all research articles. According to this format, the Methods section includes a Reagents and Tools Table (listing key reagents, experimental models, software and relevant equipment and including their sources and relevant identifiers) followed by a Methods and Protocols section describing the methods using a step-by-step protocol format. The aim is to facilitate adoption of the methodologies across labs. More information on how to adhere to this format as well as a downloadable template (.docx) for the Reagents and Tools Table can be found in our author guidelines:

13) As part of the EMBO publication's Transparent Editorial Process, EMBO Reports publishes online a Review Process File to accompany accepted manuscripts. This File will be published in conjunction with your paper and will include the referee reports, your point-by-point response and all pertinent correspondence relating to the manuscript.

Kind regards,

Martina

Referee #1:

This study builds on previous work by the Riemer lab (Finger et al. 2020) who identified cytoplasmic AK2 degradation following DPP9 cleavage. Here, the authors investigate the underlying mechanism and the intricate interplay between protein N-terminal processing events and stability within cells. The authors found that following N-terminal processing of AK2 by MAPs and DPP9, a hidden IBM is exposed. The exposed IBM on processed AK2 allows binding to IAPs like BIRC2, BIRC3, BIRC6, and XIAP. This interaction targets AK2 for degradation by the ubiquitin-proteasome system, preventing its cytoplasmic accumulation. Consistent with Mueller et al. 2021, the study suggests that acetylation of the processed AK2 N-terminus masks the IBM, preventing interaction with IAPs and subsequent degradation. Finally, authors suggested that DPP8/9-dependent unmasking of

IBMs might be a more general phenomenon. Bioinformatic analysis revealed a significant number of human proteins with potential IBMs located downstream of DPP8/9 cleavage sites suggesting a potential regulatory pathway for protein stability involving these enzymes and IAPs. To explore this broader role, the authors investigated EIF2A and found evidence suggesting it is targeted to IAPs after IBM exposure by DPP8/9.

Overall, this study utilizes well-executed techniques, generating clear and valuable data. It is fair to say the study is of interest to colleagues in the ubiquitin-proteasome field. To fully elucidate the connection between acetylation, DPP9 cleavage, IBM exposure, and IAP-mediated degradation, further exploration is warranted. While the data on EIF2A is promising, additional investigation would strengthen these findings. To be suitable for publication further the following concerns need to be addressed.

Major:

1. To strengthen the data on N-terminal processing in Fig. 1F-G, the assay should be repeated with DPP9 KO cells and in addition using WT AK2 instead of C232S. This will shed light on the proportion of processing by DPP9, comparing WT and C232S AK2.
2. The study lacks a direct demonstration of AK2 degradation by IAPs *in vivo*. While silencing all IAPs can be challenging, the chosen P6W mutant for the IBM wasn't directly tested for IAP binding. The authors rely on docking studies suggesting P6W doesn't bind XIAP's IBM-binding cleft (data not shown). Ideally, this data should be presented, or binding assay between XIAP and the P6W mutant should be performed for stronger evidence. In addition, the P6W mutant migrates faster in SDS-PAGE, potentially indicating a PTM alteration on AK2 that might indirectly affect stability. Alternatively, the authors could monitor protein levels of the previously identified "VVPL" mutant, which lacked IAP interaction (Fig. 2D, H), to strengthen the link between IAPs and AK2 degradation.
3. Fig. 4C demonstrates that silencing NatA reduces AK2 protein levels. However, the study doesn't directly address whether this reduction is dependent on IAP-mediated degradation. The ideal experiment would be to deplete IAPs in cells with silenced NatA, but, again, this can be challenging. As an alternative to depleting IAPs, a proteasome inhibitor can be added to cells with silenced NatA. This would provide evidence for whether the observed decrease in AK2 protein levels is due to proteasome-mediated degradation.
4. Fig. 5G suggests that EIF2A might not be a very efficient DPP9 substrate due to the significantly lower levels of processed peptides (STPLLTVR) compared to unprocessed forms (APSTPLLTVR) (3 orders lower). This raises questions about the extent to which DPP9 regulates EIF2A stability. To better evaluate the impact of DPP9 KO on EIF2A protein levels, samples in Fig. 5H should be run on the same gel for a direct comparison. In addition, the lack of a clear difference in EIF2A protein levels upon IAP/BIRC2 silencing (Fig. 5H) warrants further investigation. This leaves the dependency of EIF2A stability on IAPs unclear, and to what extent ubiquitination differences translate to turnover rate variations remains uncertain. Lastly, the potential regulation of EIF2A by NatA is not explored. Monitoring protein levels of EIF2A in NatA-depleted cells could provide valuable insights.

Minor:

5. N-terminal acetylation is believed to occur co-translationally. The data suggests that AK2 acetylation take place post-translationally. Can authors comment on what is the proportion of acetylation of DPP9-process AK2 vs. non acetylated species?
6. Given the loose consensus sequence of IBM motifs, 129 proteins were suggested to have DPP8/9 + IBM motifs. It will be informative to cross this dataset with publicly available datasets of known DPP8/9 substrates, such as those described in Wilson et al. (2013), Zhang et al. (2015), and Shimshon et al. (2024). This data-driven approach would leverage existing knowledge to assess the likelihood that these newly identified proteins with IBM motifs are indeed regulated by DPP8/9.
7. Page 5: "the assay indicated that ca 80% of cellular AK2...". Delete "ca"?
8. Fig. 2A- To clarify the consensus in the first four residues of IBM motifs, it would be beneficial to include numbering (1-4) beneath the four columns.
9. Fig. 4A- BIRC2 appears twice.
10. While the manuscript describes peptide pulldown experiments, including the mass spectrometry data as supplemental spreadsheets would be beneficial for readers. This would allow researchers to further analyze the data for protein interactions.
11. Page 8- "acetylated protein N-termini starting with serine or alanine were proposed to act as potential degradation motifs (Ac-N- degrons) (Hwang et al, 2010)."
The statement concerning N-terminal acetylation and protein turnover in the referenced paper requires correction. The paper focuses on the impact of N-terminal acetylation on protein turnover within yeast. The specific focus lies on proteins where the N-terminal sequence begins with MN or ML. The study does not analyze the acetylation of proteins with N-terminal sequences starting with alanine or serine. In fact, the peptides analyzed in this context had N-terminal small amino acid followed by leucine (AL, SL, CL, VL, TL) and their turnover was assayed in yeast. The data presented does not imply a direct link between N-terminal acetylation and the observed protein degradation.

Referee #2:

The authors present an interesting study on substrate cleavage by cytosolic DPPs and the functional consequence(s) of

substrate truncation; here for the protein adenylate kinase 2 (AK2).

They convincingly show N-terminal processing of AK2 by DPPs 8 and 9, exposing a novel degradation motif that is recognized by the IAP protein family of ubiquitin ligases.

Further, they highlight the stabilizing role of N-terminal acetylation of AK2.

Lastly, they highlight the likely presence of this DPP-type regulation in additional cytosolic proteins.

This is an interesting study. The presented data nicely backs the conclusions of the authors. There seems to be no over-interpretation of the data.

Comments and questions:

Page 5: The authors write that AK2 is acetylated after (sic!) removal of (a) the initiator methionine (by MAPs) and (b) removal of the neo N-terminal AP by DPPs. To my understanding, N-terminal acetylation is largely a co-translational event. Could the authors please comment on whether this would implicate DPPs being present at the translational machinery, similar to MAPs?

The authors write that "removal of only two amino acids is difficult to monitor". However, the AK2 N-terminus is readily accessible to mass spectrometry with a suitably sized peptide following trypsinization or cleavage at R (in case of chemically modified K). I suggest that the authors investigate the various proteomics and N-terminomics databases, such as TopFind and PeptideAtlas for N-terminal peptides of AK2.

Do the authors suspect further N-terminal trimming of AK2, e.g. after the second or third proline? Could the authors comment on how to distinguish these processing steps and how they verified the actual N-terminal sequence of (processed) AK2?

Fig 1F: a molecular weight marker would be helpful. Or is this a schematic western blot?

The peptide pull-down experiments (incl. the acetylated peptides) are very interesting and nicely performed. I would suggest to include more information into this results section, e.g. number of replicates, coverage, etc.

The authors conclude their study by demonstrating the wider significance of their findings by a genome-wide sequence motif search. DPP8/9 inhibitors have been developed and are being published. Could the authors speculate (or even add data) on how DPP8 inhibition affects this intriguing pathway?

Referee #3:

The manuscript proposes a mechanism that could enable proteasomal degradation of AK2 and prevent aberrant cytosolic activity of this protein. Previously published work demonstrated that AK2 is processed by DPP8/9 in route to the mitochondria. Cytosolic, DPP8/9-processed AK2 is then degraded by the proteasome. In this new manuscript, the authors reason that AK2 processing by DPP8/9 reveals an N-terminal region that might function as an IAP(inhibitors of apoptosis)-binding motif (IBM). The main line of evidence of this work is the use of peptides containing the N-terminal region of DPP8/9-processed and unprocessed AK2. These peptides are used in pull-down experiments for identification of interacting proteins. Among the main hits are members of the IAP family. A genomic search identifies more than 100 proteins that may be subject to DPP8/9 cleavage and contain potential IBMs adjacent to the cleavage site. The authors then use a similar peptide-based strategy to demonstrate that one of the identified proteins (EIF2A) might also be subject to the proposed mechanism. This suggests that DPP8/9 processing might be a general mechanism that exposes IBMs in proteins. While the peptide pull-down/MS data is coherent and is supported by adequate controls, this reviewer considers that alternative experiments are not as strong and need further assessment. Whereas the work is potentially interesting to a wide community, certain aspects lack rigor and thus diminish any impact on the field.

Main comments:

1. The main pitfall relates to the lack of demonstration of the proposed mechanism by experiments alternative to peptide pull-downs. AK2 is a known DPP8 substrate and processing enables proteasomal degradation (Finger, 2020). The novelty of this work relies on the intermediate steps, where DPP8/9 processing enables interaction of AK2 with IAPs, which may ubiquitinate AK2 causing protein degradation. While the evidence using peptide pull-downs is good, this reviewer suggests that stronger cellular and biochemical strategies should be completed to support the model.

2. A major concern relies on the coprecipitation of AK2 and the potentially interacting IAPs.

For example, in Figure 2M, the western blot showing XIAP coprecipitation with AK2 is far from solid. The difference is not clear when comparing induced vs non-induced cells. Quantification with replicates would be required at this point. In addition, coprecipitation of a single protein (XIAP) is insufficient for this demonstration, given that other IAPs were identified by peptide pull-down. Similar experiments to those shown in Figure 2L-M with other identified IAPs would enrich the manuscript. Moreover, the use appropriate controls, including coprecipitation with unprocessed AK2-HA and a known IAP binder would be appropriate here.

3. Given that IAPs are multi-domain proteins with different binding partners and functions, this reviewer argues that more evidence is necessary to propose an association between binding of DPP8/9-processed AK2 and IAPs, to the consequential ubiquitination and degradation of AK2. In other words, IAPs might be AK2 binding partners but not necessarily induce ubiquitination and degradation of AK2. I suggest in vitro ubiquitination assays of recombinant AK2 and using the ubiquitin ligases discovered by peptide pull-downs. Alternatively, similar experiments to those shown in Figure 5H could be done with AK2.

Minor comments:

1. Figure 1B-E and 1G: Please specify what cell line was used for stable expression of AK2 variants.
2. Figures 1E, 3C: Please define your western blot labels (T, C, M).
3. Figure 2M: Lane labeling 1-4 is not clear. Same labeling is found in other figures, but numbers are not specified.
4. Please correct the following to a more accurate statement: "We thereby observed only a minor but not significant stabilization of cytosolically confined AK2-3CS (Figure 3A)." Neither the western blot nor the graph shows a "minor stabilization of AK2".
5. In figure 3A, AK2 stabilization experiments are done on a 3CS background. Then, a WT background was used for figure 3B and 3C. Please clarify what is the rationale for this change?
6. Visual inspection of Figure 3C indicates an accumulation of AK2 in the mitochondrial fraction, but the authors solely focus on the cytoplasmic fraction. Please take this result into account.
7. As shown in Figure 1E, AK2-HA WT is expected to be present in the mitochondrial fraction. However, in figure 3C this effect is no longer observed. How do the authors account for this discrepancy?
8. In figure 3A, what was used for normalization of HA signal?

Point-by-Point Response**Lapacz et al - DPP8/9 processing of human AK2 unmask an IAP binding motif**

We thank the referees for the positive assessment of our study. We addressed all referee comments most of them experimentally. We added further evidence on the N-terminal processing of AK2 (**new Figure 1F**), we demonstrated its ubiquitination in an *in vitro* ubiquitination assay (**new Figure 3I**). We also demonstrated the interaction between AK2 and IAPs using complementing methods (**new Figures 2K-N**). Moreover, we demonstrated for EIF2A the presence of DPP8/9 processed N-termini in intact cells (also with N-acetylations), showed ubiquitination in *in vitro* ubiquitination assays, and provided data that acetylation introduced by NatA increased stability (**new panels in Figure 4**).

We are convinced that our new data greatly improved our study convincingly demonstrating that DPP8/9 can unmask previously hidden IBMs in various proteins thereby adding a new layer of regulation to protein degradation.

Referee #1:

This study builds on previous work by the Riemer lab (Finger et al. 2020) who identified cytoplasmic AK2 degradation following DPP9 cleavage. Here, the authors investigate the underlying mechanism and the intricate interplay between protein N-terminal processing events and stability within cells. The authors found that following N-terminal processing of AK2 by MAPs and DPP9, a hidden IBM is exposed. The exposed IBM on processed AK2 allows binding to IAPs like BIRC2, BIRC3, BIRC6, and XIAP. This interaction targets AK2 for degradation by the ubiquitin-proteasome system, preventing its cytoplasmic accumulation. Consistent with Mueller et al. 2021, the study suggests that acetylation of the processed AK2 N-terminus masks the IBM, preventing interaction with IAPs and subsequent degradation. Finally, authors suggested that DPP8/9-dependent unmasking of IBMs might be a more general phenomenon. Bioinformatic analysis revealed a significant number of human proteins with potential IBMs located downstream of DPP8/9 cleavage sites suggesting a potential regulatory pathway for protein stability involving these enzymes and IAPs. To explore this broader role, the authors investigated EIF2A and found evidence suggesting it is targeted to IAPs after IBM exposure by DPP8/9.

Overall, this study utilizes well-executed techniques, generating clear and valuable data. It is fair to say the study is of interest to colleagues in the ubiquitin-proteasome field. To fully elucidate the connection between acetylation, DPP9 cleavage, IBM exposure, and IAP-mediated degradation, further exploration is warranted. While the data on EIF2A is promising, additional investigation would strengthen these findings. To be suitable for publication further the following concerns need to be addressed.

We thank this referee for the positive assessment of our study.

Major:

1. To strengthen the data on N-terminal processing in Fig. 1F-G, the assay should be repeated with DPP9 KO cells and in addition using WT AK2 instead of C232S. This will shed light on the proportion of processing by DPP9, comparing WT and C232S AK2.

Answer:

We repeated the experiment using the competitive inhibitor 1G244 in cells expressing AK2 C232S or AK2 C232S,A2C. Both variants can be shifted upon reduction and mmPEG12 addition, in line with the presence of the cysteines C40, C42 and C92. However, only the A2C variant (containing an additional cysteine) shows a further band appearing on top of the shifted band. This band corresponds to DPP8/9

unprocessed AK2. Upon 1G244 treatment AK2 signals become consistently more intense (**new Figure S2B, lanes 2,4,6,8**). When comparing the A2C variant with and without 1G244 we could however only observe a minimal effect on processing (**new Figure S2B, quantification**). We thus decided to add proteomics data assessing N-terminal processing of endogenous AK2 in WT and DPP9 KO cells (**new Figure 1F**). These revealed that in DPP9 KO cells only a third of the processed N-terminal peptides could be found compared to the WT situation.

We prepared the A2C variant only in the background of the C232S mutant and could thus not repeat the experiment with WT AK2.

2. The study lacks a direct demonstration of AK2 degradation by IAPs in vivo. While silencing all IAPs can be challenging, the chosen P6W mutant for the IBM wasn't directly tested for IAP binding. The authors rely on docking studies suggesting P6W doesn't bind XIAP's IBM-binding cleft (data not shown). Ideally, this data should be presented, or binding assay between XIAP and the P6W mutant should be performed for stronger evidence. In addition, the P6W mutant migrates faster in SDS-PAGE, potentially indicating a PTM alteration on AK2 that might indirectly affect stability. Alternatively, the authors could monitor protein levels of the previously identified "VVPL" mutant, which lacked IAP interaction (Fig. 2D, H), to strengthen the link between IAPs and AK2 degradation.

Answer:

In this revised manuscript version, we now in detail explain the principles of IAP binding to the AK2 IBM and the unstructured quality of the AK2 N terminus and provide updated figures (**new text below and new Figures 2B,C**). Modelling of AK2 P6W with XIAP did not yield any “meaningful” result (*i.e.* there is simply no binding observable or the sequence is modelled into the IBM binding cleft with an offset– therefore we did not show the data. We removed the “data not shown” part of the text.

New text to Figures 2B,C:

“Indeed, a structure prediction using AlphaFold 3 (Abramson et al, 2024) showed binding of the AK2 “SVPA”-N-terminal motif to the BIR3-domain of XIAP with high confidence, consistent with the reported structure of the similar “AVPI” tetrapeptide of the N-terminus of SMAC/DIABLO bound to the XIAP BIR3 domain (Wu et al., 2000) (Figure 2B). Systematic studies have shown that “SVPI” tetrapeptides can bind either the BIR2 and BIR3 domains with a preference for the BIR3 domain, although with slightly lower affinity than “AVPI” tetrapeptides (Lukacs et al, 2013). At the P4 position of the peptide, alanine is tolerated similarly to isoleucine (Lukacs et al., 2013; Sweeney et al, 2006), supporting the binding capability of the processed AK2 N-terminus. Additionally, the N-terminal 15 amino acids of AK2 are predicted to be disordered and exposed, granting accessibility of folded AK2 to IAPs (Figure 2C).”

New text to Figures 3B,C

“It has been shown that at the P3 position of IBMs (corresponding to AK2 P6), proline is conserved, while tryptophan is highly disfavoured and disrupts IAP binding, in particular to BIR3 of XIAP (Lukacs et al., 2013; Sweeney et al., 2006). In agreement with this, an AK2-P6W variant was present at higher levels when compared to AK2 wildtype indicating a decreased degradation of the protein in the cytosol (Figure 3B). In line with a decreased cytosolic degradation, AK2-P6W in part accumulated in the cytosolic fraction (Figure 3C). Likewise, another AK2 variant bearing the S4V mutation that can also not interact with IAPs ((Lukacs et al., 2013; Sweeney et al., 2006), Figure 2H) was stabilized (Figure S5, (Finger et al., 2020)), collectively demonstrating that an intact IBM determines AK2 stability in cells.”

We also report now on the (increased) cellular levels of the AK2 S4V mutant (**new Figure S5**) that did not bind IAPs (Figure 2H). We had previously already used such a mutant and found it to be stabilized

(Finger, Habich et al, EMBO J 2020). In the context of this study, we focused on the P6W mutant to not change the N-terminal amino acid residues of AK2 after DPP8/9 processing.

We also included data that confirm IAP binding to AK2 in intact cells using orthogonal experiments (**new Figures 2K-N**) and we show in an *in vitro* ubiquitination experiment that AK2 can be ubiquitinated by XIAP but only if it presents its free and DPP8/9 processed N-terminus (**new Figure 3H,I**).

3. Fig. 4C demonstrates that silencing NatA reduces AK2 protein levels. However, the study doesn't directly address whether this reduction is dependent on IAP-mediated degradation. The ideal experiment would be to deplete IAPs in cells with silenced NatA, but, again, this can be challenging. As an alternative to depleting IAPs, a proteasome inhibitor can be added to cells with silenced NatA. This would provide evidence for whether the observed decrease in AK2 protein levels is due to proteasome-mediated degradation.

Answer:

We now added an experiment in which we assessed AK2 3CS levels upon NatA depletion in combination with proteasome inhibition (**new Figure 3G**). We confirmed again the slight reduction in AK2 3CS levels upon NatA depletion. We also demonstrated stabilization of AK2 3CS by MG132 indicating proteasomal degradation of the protein.

4. Fig. 5G suggests that EIF2A might not be a very efficient DPP9 substrate due to the significantly lower levels of processed peptides (STPLLTVR) compared to unprocessed forms (APSTPLLTVR) (3 orders lower). This raises questions about the extent to which DPP9 regulates EIF2A stability. To better evaluate the impact of DPP9 KO on EIF2A protein levels, samples in Fig. 5H should be run on the same gel for a direct comparison.

In addition, the lack of a clear difference in EIF2A protein levels upon IAP/BIRC2 silencing (Fig. 5H) warrants further investigation. This leaves the dependency of EIF2A stability on IAPs unclear, and to what extent ubiquitination differences translate to turnover rate variations remains uncertain.

Lastly, the potential regulation of EIF2A by NatA is not explored. Monitoring protein levels of EIF2A in NatA-depleted cells could provide valuable insights.

Answer:

As requested by this referee, we performed additional experiments to provide further evidence for the processing, ubiquitination and degradation of EIF2A.

We reanalysed the proteomics data of the EIF2A IPs treated with MG132 with a modified human FASTA file, where proteins starting with MP or MYP (Y = any amino acid) were modified by removing MP or MYP while others remained in full-length. Nt-acetylation was set as a variable modification. Our prior analysis allowed cleavage after proline in the data analysis (in addition to lysine and arginine after tryptic digest) to detect cleaved peptides but did not detect N-terminal acetylation for cleaved peptides since these peptides were not treated as N-terminal. With the modified database, cleaved peptides were recognized as N-terminal, enabling acetylation detection. We identified both cleaved acetylated and free cleaved peptides in WT and DPP9 KO HEK cell lines of EIF2A. We thereby found that in DPP9 KO cells, levels of the processed EIF2A-peptide drop to below 30% of the levels in WT cells (**new Figures 4C,D**), and that about 30% of processed DPP9-processed N-termini might be acetylated (see also answer below). We want to mention here that in our previous analysis the cleaved peptide intensity was around 30x lower in the WT and not detected in the DPP8/9 KO. The discrepancy between the numbers presented in this revised manuscript version and the original submission might be explained as follows: We used for the first submission DIA-NN version 1.8.2. beta and allowed

cleavage after proline to identify the EIF2A cleaved N-terminal peptide. In the revised manuscript, we used the new release of DIA-NN (version 1.9.1) and the modified FASTA file to identify the cleaved peptide which resulted in higher intensities for the cleaved peptide. Allowing cleavage after proline increases the search space many times which might explain differences. After having noticed this discrepancy, we analysed the data with DIA-NN version 1.8.2 beta, 1.9.2. (newest release) and MaxQuant DIA (v2.6.7.0) with the modified database and a human FASTA file only modified for MAP of EIF2A. We identified the cleaved peptide always with higher intensities as in the first submission and decided to report the values of the new analysis. In addition, we inspected several MS/MS spectra to be sure about the correct peptide identification. Despite this difference in intensity, it is important to mention that intensities of different peptides cannot be directly compared based on our MS analysis due to differing peptide properties. Additional laborious experiments e.g. spike-in MS experiments would be necessary to be able to make solid conclusions about cleavage proportions.

To expand the insights on EIF2A processing, we also added data assessing EIF2A levels upon Nata depletion (**new Figure 4I**). We thereby observed a moderate decrease in EIF2A levels upon Nata depletion. We further performed an *in vitro* ubiquitination experiment of EIF2A peptides by XIAP demonstrating that only upon exhibition of its DPP8/9-processed free N-terminus it can be ubiquitinated (**new Figure 4J**)

Lastly, we provide the gel depicted in **Figure 4K** (previously Figure 5H) on one gel. This clearly demonstrates that XIAP/BIRC2-dependent ubiquitination is DPP8/9 processing dependent. Diminished processing in DPP9 KO cells still results in ubiquitination of EIF2A; it is however not dependent on XIAP/BIRC2 anymore as it does not carry an IBM at its N-terminus.

Minor:

5. N-terminal acetylation is believed to occur co-translationally. The data suggests that AK2 acetylation take place post-translationally. Can authors comment on what is the proportion of acetylation of DPP9-process AK2 vs. non acetylated species?

Answer:

We appreciate the reviewer's comment and agree that N-terminal acetylation by Nata typically occurs co-translationally. To address this, we reanalyzed our MS dataset of HEK WT and DPP9 KO proteomes (13,262 quantified proteins) using a modified human FASTA file (as explained above), where proteins starting with MP or MYP (Y = any amino acid) were modified by removing MP or MPX, while others remained in full-length. Nt-acetylation was set as a variable modification. This analysis identified 2,668 N-terminal peptides, including 26 MS/MS spectra of DPP9-cleaved targets. Notably, UGGT2 was found with an N-terminal acetyl group (MS/MS spectrum attached). Attempts to identify acetylated or unacetylated (for UGGT2) counterparts were unsuccessful, but this finding, along with previous identification of the cleaved, acetylated peptide of AK2 (Finger et al., EMBO J., 2020), supports post-cleavage N-terminal acetylation by DPP8/9.

We also reanalyzed our EIF2A immunoprecipitation MS data using the same modified human FASTA file (see answer above). Our prior analysis allowed cleavage after P in DIA-NN but did not detect N-terminal acetylation since these peptides were not treated as N-terminal. With the modified database, cleaved peptides were recognized as N-terminal, enabling acetylation detection. We identified both cleaved acetylated and free cleaved peptides in WT and DPP9 KO HEK cell lines. Due to differing peptide properties, direct intensity comparisons were not feasible. Thus, we calculated the ratio of the acetylated and free cleaved peptide to all measured unmodified EIF2A internal peptides (39 peptides). The calculated ratios were as follows:

	Ratio: free cleaved peptide/all EIF2A internal peptides	Ratio: acetylated cleaved peptide/all EIF2A internal peptides	Ratio of ratio (free/acetylated)
WT HEK	0.5	0.16192	3.09
DPP8/9 KO HEK	0.1047	0.03718	2.82

These results suggest that approximately 1/3 of the cleaved N-terminus of EIF2A is acetylated, though precise quantification would require further spike-in experiments.

In conclusion, we have identified three cases (AK2, EIF2A, UGGT2) demonstrating cleaved, acetylated N-termini, with EIF2A data suggesting roughly 1/3 acetylation. This raises intriguing questions regarding whether DPP9 may act at the ribosome, facilitating co-translational acetylation, or if an unidentified enzyme or NatA mediates this process post-translationally.

6. Given the loose consensus sequence of IBM motifs, 129 proteins were suggested to have DPP8/9 + IBM motifs. It will be informative to cross this dataset with publicly available datasets of known DPP8/9 substrates, such as those described in Wilson et al. (2013), Zhang et al. (2015), and Shimshon et al. (2024). This data-driven approach would leverage existing knowledge to assess the likelihood that these newly identified proteins with IBM motifs are indeed regulated by DPP8/9.

We have performed such a cross-referencing analysis with the literature and incorporated its results into the main text (red):

“Lastly, we found a subset of 129 proteins that contain DPP8/9-cleavage site-masked IBMs. Among them are besides AK2 and EIF2A, many metabolic enzymes. DPP8/9 have been linked to metabolic rewiring (Finger et al., 2020; Wilson et al., 2013; Zhang et al, 2015), and it will be exciting to test in the future whether these proteins are DPP8/9 targets and whether IBM exposure serves regulatory purposes during differentiation and shifts in metabolism. By comparing our list to results of DPP8/9-substrate screens in the literature (Wilson et al., 2013; Zhang et al, 2015), we further noted that the protein encoded by S100A10 is a validated substrate of DPP9. Furthermore, there is experimental evidence supporting the protein disulfide-isomerase encoded by TXNDC5 and proteins of the semaphorin family (encoded by mouse genes Plxn1 and Sema6c, whereas in humans, we detected PLXNA3 and SEMA4F) as substrates.”

In addition, we analysed available unpublished datasets from our labs (see screenshot below) and identified AK2, EIF2A, S100A10, SF3A1, ASCC2, RCOR1 and POMT2 from our list in the DPP8/9 cleaved state further strengthening that the targets are indeed processed.

uniprot_id	gene_name	start_peptide_position	cleavage_after_position	site-3	site-2	site-1	site+1	site+2	site+3	MAP_site	MP_site	DPP9_cleavage_motif
704263 P54819	AK2	4	3	M	A	P	S	V	P	True	False	True
734704 P60903	S100A10	3	2	_	M	P	S	Q	M	False	True	True
975118 Q15459	SF3A1	3	2	_	M	P	A	G	P	False	True	True
1268056 Q9BY44	EIF2A	4	3	M	A	P	S	T	P	True	False	True
1283917 Q9H118	ASCC2	3	2	_	M	P	A	L	P	False	True	True
1385008 Q9UKL0	RCOR1	469	468	K	M	P	E	E	E	False	True	True
1387127 Q9UKY4	POMT2	3	2	_	M	P	P	A	T	False	True	True
1457530 P08207	S100A10	3	2	_	M	P	S	Q	M	False	True	True
1504793 Q9WTP6	AK2	4	3	M	A	P	N	V	L	True	False	True

7. Page 5: "the assay indicated that ca 80% of cellular AK2...". Delete "ca"?

Answer:

Done.

8. Fig. 2A- To clarify the consensus in the first four residues of IBM motifs, it would be beneficial to include numbering (1-4) beneath the four columns.

Answer:

We added the numbering of the IBM to the panel.

9. Fig. 4A- BIRC2 appears twice.

Answer:

There are two entries in the uniprot human FASTA file (Q13490 and E9PMH5). E9PMH5 is missing amino acid 1-22 from the full-length Q13490. We apologize that we did not notice that both variants were reported and marked now only the full-length BIRC2 in the figure.

10. While the manuscript describes peptide pulldown experiments, including the mass spectrometry data as supplemental spreadsheets would be beneficial for readers. This would allow researchers to further analyze the data for protein interactions.

Answer:

We agree with the reviewer and included now all MS data also as supplementary excel sheets.

11. Page 8- "acetylated protein N-termini starting with serine or alanine were proposed to act as potential degradation motifs (Ac-N- degrons) (Hwang et al, 2010)."

The statement concerning N-terminal acetylation and protein turnover in the referenced paper requires correction. The paper focuses on the impact of N-terminal acetylation on protein turnover within yeast. The specific focus lies on proteins where the N-terminal sequence begins with MN or ML. The study does not analyze the acetylation of proteins with N-terminal sequences starting with alanine or serine. In fact, the peptides analyzed in this context had N-terminal small amino acid followed by leucine (AL, SL, CL, VL, TL) and their turnover was assayed in yeast. The data presented does not imply a direct link between N-terminal acetylation and the observed protein degradation.

Answer:

The referenced study shows these small aa as potential degrons when acetylated and provides spot assays (Fig 2E in the paper) showing strongly enhanced interaction of Doa10 with acetylated N-termini over non-acetylated, and states: "N-terminal Met, Ala, Val, Ser, Thr, and Cys are shown here to function as secondary destabilizing residues in the N-end-rule pathway, in that they must be Nt-acetylated before their recognition by the *S. cerevisiae* Doa10 Ub ligase as N-degrons, termed AcN-degrons, that require Nt-acetylation (Fig. 4)."

Indeed, the residues are followed by Leucine, which is not V or T as in AK2 or EIF2A, but the focus is still on the potential to function as degron.

Thus, we respectfully disagree here with the referee. However, to avoid potential confusion with readers, we removed the sentence from the manuscript.

Referee #2:

The authors present an interesting study on substrate cleavage by cytosolic DPPs and the functional consequence(s) of substrate truncation; here for the protein adenylate kinase 2 (AK2).

They convincingly show N-terminal processing of AK2 by DPPs 8 and 9, exposing a novel degradation motif that is recognized by the IAP protein family of ubiquitin ligases. Further, they highlight the stabilizing role of N-terminal acetylation of AK2. Lastly, they highlight the likely presence of this DPP-type regulation in additional cytosolic proteins. This is an interesting study. The presented data nicely backs the conclusions of the authors. There seems to be no over-interpretation of the data.

We thank this referee for the positive assessment of our study.

Comments and questions:

Page 5: The authors write that AK2 is acetylated after (sic!) removal of (a) the initiator methionine (by MAPs) and (b) removal of the neo N-terminal AP by DPPs. To my understanding, N-terminal acetylation is largely a co-translational event. Could the authors please comment on whether this would implicate DPPs being present at the translational machinery, similar to MAPs?

Answer:

We agree with this referee that our data implicate a co-translational processing of proteins by DPP8/9 if the N-terminal acetylation would always be cotranslational. Further analysis of our proteomics data also demonstrated acetylation of EIF2A at its DPP8/9-processed N-terminus (**new Figure 4D, second plot**) emphasizing the occurrence of this modification (see as well answer to the reviewer 1 point 5).

In the framework of this revision, we also performed native IP experiments of RPL22-HA (a subunit of the cytosolic ribosome, see volcano plot) in the hope to identify coprecipitating DPPs. While we identified a NAT (as example of a ribosomal interactor) and RPL22, we failed to detect any DPP, either because the DPP amounts associated with the ribosome were too low, the interaction was too transient, or there was no interaction, and acetylation of processed N-termini can also occur post-translationally.

The authors write that "removal of only two amino acids is difficult to monitor". However, the AK2 N-terminus is readily accessible to mass spectrometry with a suitably sized peptide following trypsinization or cleavage at R (in case of chemically modified K). I suggest that the authors investigate the various proteomics and N-terminomics databases, such as TopFind and PeptideAtlas for N-terminal peptides of AK2.

Answer:

We thank the reviewer for this suggestion. We reanalyzed our MS HEK WT and HEK DPP9 KO proteomes using a modified human FASTA file as explained above (answer to reviewer 1 point 5). In this version, proteins starting with MP or MXP (X = any amino acid) were adjusted by removing MP or MYP, while others remained in full-length to monitor cleavage of AK2 and other potential targets. Through this analysis, we identified several N-terminal versions of AK2, summarized in **new Figure 1F**. MS/MS spectra are provided in **new Figure S1**, and all identified AK2 peptides are listed in a supporting excel file (Table EV1). Additionally, we searched the N-terminomics databases TopFIND and PeptideAtlas for AK2-cleaved peptides. Identified N-termini include positions 1 (unprocessed), 2 (methionine-cleaved), 4 (DPP9 cleavage), and 7 (potential second cleavage site), with additional peptides starting from positions 5 and 8 also reported.

Do the authors suspect further N-terminal trimming of AK2, e.g. after the second or third proline?
Could the authors comment on how to distinguish these processing steps and how they verified the actual N-terminal sequence of (processed) AK2?

Answer:

To investigate second or third cleavage events, we applied a similar MS database search as above to our MS HEK WT and DPP9 KO proteome data set. We used a human FASTA file, where we deleted once the first 6 and then the first 9 aa of AK2 to be able to identify possible N-terminal versions of additional cleavages. This approach did not identify additional cleaved versions of AK2. We performed the same approach for the MS immunoprecipitation data of EIF2A which has as well another proline at position 6 with the same result that we did not detect further cleaved N-termini. For identified N-terminal versions of AK2 and EIF2A from our searches we extracted the MS/MS spectra to verify the identified sequences (acM unprocessed, Methionine cleaved AP starting and DPP9 cleaved for AK2; M unprocessed, Methionine cleaved AP starting and DPP8/9 cleaved for the acetylated and free version of EIF2A; the MS/MS spectra can be found in the **new Figure S8**). N-terminomics databases indicate that cleavage after the second proline of AK2 (Topfind and PeptideAtlas) and EIF2A (PeptideAtlas) exist and we cannot exclude further N-terminal processing. We want to point out that we concentrated in this work on DPP8/9 cleavage sites which expose an IAP-binding motif and IAP recognition through the cleavage which is only the case for the first cleavage site of AK2 and EIF2A.

Fig 1F: a molecular weight marker would be helpful. Or is this a schematic western blot?

Answer:

This was a scheme. The panels are now moved to the SI.

The peptide pull-down experiments (incl. the acetylated peptides) are very interesting and nicely performed. I would suggest to include more information into this results section, e.g. number of replicates, coverage, etc.

Answer:

We thank the reviewer for this comment. We felt that adding additional information and numbers in the main text did disturb the easy reading of the text and added now some additional information in the figure legend and material methods part. We hope that the reviewer agrees with this. Additionally, we provide now all excel tables with the t-test data for more information.

The authors conclude their study by demonstrating the wider significance of their findings by a genome-wide sequence motif search. DPP8/9 inhibitors have been developed and are being published. Could the authors speculate (or even add data) on how DPP8 inhibition affects this intriguing pathway?

Answer:

Gene expression data show distinct expression patterns of DPP8 and 9. In the HEK293 cells used here, DPP9 appears to be the dominant peptidase. We assume that regulation of both DPP8 and 9 expression levels and activity (e.g. through PTMs) can impact this pathway and might serve in regulating the stability of the identified targets. We speculate that DPP8 and 9 take here tissue- and development/differentiation-specific roles.

Referee #3:

The manuscript proposes a mechanism that could enable proteasomal degradation of AK2 and prevent aberrant cytosolic activity of this protein. Previously published work demonstrated that AK2 is processed by DPP8/9 in route to the mitochondria. Cytosolic, DPP8/9-processed AK2 is then degraded by the proteasome. In this new manuscript, the authors reason that AK2 processing by DPP8/9 reveals an N-terminal region that might function as an IAP (inhibitors of apoptosis)-binding motif (IBM). The main line of evidence of this work is the use of peptides containing the N-terminal region of DPP8/9-processed and unprocessed AK2. These peptides are used in pull-down experiments for identification of interacting proteins. Among the main hits are members of the IAP family. A genomic search identifies more than 100 proteins that may be subject to DPP8/9 cleavage and contain potential IBMs adjacent to the cleavage site. The authors then use a similar peptide-based strategy to demonstrate that one of the identified proteins (EIF2A) might also be subject to the proposed mechanism. This suggests that DPP8/9 processing might be a general mechanism that exposes IBMs in proteins. While the peptide pull-down/MS data is coherent and is supported by adequate controls, this reviewer considers that alternative experiments are not as strong and need further assessment. Whereas the work is potentially interesting to a wide community, certain aspects lack rigor and thus diminish any impact on the field.

Main comments:

1. The main pitfall relates to the lack of demonstration of the proposed mechanism by experiments alternative to peptide pull-downs. AK2 is a known DPP8 substrate and processing enables proteasomal degradation (Finger, 2020). The novelty of this work relies on the intermediate steps, where DPP8/9 processing enables interaction of AK2 with IAPs, which may ubiquitinate AK2 causing protein degradation. While the evidence using peptide pull-downs is good, this reviewer suggests that stronger cellular and biochemical strategies should be completed to support the model.

Answer:

We thank the referee for acknowledging the novelty of our study and appreciating the quality of our peptide pull down data. We strengthened these data now by:

1. ...providing proteomics data on N-termini of endogenous AK2 and EIF2A in intact cells. These data support the processing by DPP8/9 (**new Figures 1F,4D**)
2. ...providing *in vitro* ubiquitination assay data for AK2 and EIF2A. N-terminal peptides of both proteins can be ubiquitinated by purified XIAP but only if the peptide represents the free (i.e. non-acetylated) DPP8/9 processed N-terminus of both proteins (**new Figures 3I and 4J**)
3. ...providing further data demonstrating the interaction of AK2 with IAPs. We performed an IP-MS experiment to detect IAPs interacting with AK2 (**new Figure 2K-N**; replaced previous **Figure 2K-M**, which is now presented as **Figure S4**)
4. ...providing data on the levels of AK2 and EIF2A upon depletion of NataA (**Figures 3F, new 3G, new 4I**). These data demonstrate that N-terminal acetylation of both proteins protects from proteasomal degradation.

The previous version of the manuscript already contained data demonstrating that mutating the IBM in AK2 (= P6W mutation) stabilizes AK2. We now complement this by showing data using the S4V mutant of AK2 (**new Figure S5**) that cannot bind IAPs (**Figure 2H**). We show that this variant is also stabilized in line with our previous findings in Finger et al, EMBO J 2020. Likewise, the previous version of the manuscript already demonstrated a dependency of EIF2A ubiquitination on XIAP/BIRC2 (**Figure 4K**).

We are convinced that the additional data strengthen the evidence that DPP8/9 processing enables IAP binding, and subsequent ubiquitination and degradation of AK2 and EIF2A. We hope that this referee agrees with this notion.

2. A major concern relies on the coprecipitation of AK2 and the potentially interacting IAPs. For example, in Figure 2M, the western blot showing XIAP coprecipitation with AK2 is far from solid. The difference is not clear when comparing induced vs non-induced cells. Quantification with replicates would be required at this point. In addition, coprecipitation of a single protein (XIAP) is insufficient for this demonstration, given that other IAPs were identified by peptide pull-down. Similar experiments to those shown in Figure 2L-M with other identified IAPs would enrich the manuscript. Moreover, the use appropriate controls, including coprecipitation with unprocessed AK2-HA and a known IAP binder would be appropriate here.

Answer:

We agree with the reviewer and provide now additional data to support the interaction of full-length AK2 with XIAP and BIRC2. Immunoprecipitation experiments were performed using 3CS AK2-HA (cytoplasmic, capable of interacting with IAPs) and S4P-3CS AK2-HA (negative control with S4P mutation preventing IAP binding) as baits. MS analysis of IPs (n=3 for each variant) showed significant interaction with XIAP and BIRC2. For statistical analysis and data visualization, missing values were imputed (downshift 2, width 0.3). But notably, XIAP and BIRC2 were consistently detected in all 3 replicates of 3CS AK2-HA samples and were absent in the mock control and S4P-3CS samples, further consolidating specific interaction with 3CS cytoplasmic AK2. Considering the detection limit for quantified proteins in these MS runs, XIAP was enriched at least 81-fold and BIRC2 22-fold compared to mock and negative controls (**new Figure 3S**).

3. Given that IAPs are multi-domain proteins with different binding partners and functions, this reviewer argues that more evidence is necessary to propose an association between binding of DPP8/9-processed AK2 and IAPs, to the consequential ubiquitination and degradation of AK2. In other words, IAPs might be AK2 binding partners but not necessarily induce ubiquitination and degradation of AK2. I suggest in vitro ubiquitination assays of recombinant AK2 and using the ubiquitin ligases discovered by peptide pull-downs. Alternatively, similar experiments to those shown in Figure 5H could be done with AK2.

Answer:

As suggested by this referee, we performed *in vitro* ubiquitination assays using purified XIAP (**new Figure 3I, 4J**). We demonstrate that only peptides representing the DPP8/9 processed and free EIF2A and AK2 N-termini can be recognized and ubiquitinated by XIAP.

Minor comments:

1. Figure 1B-E and 1G: Please specify what cell line was used for stable expression of AK2 variants.

Answer:

AK2 variants were expressed in HEK293 based cell lines – the Flp-In™ T-REx™ 293 Cell Line. DPP9 KO cells were generated using CRISPR/Cas9 technology. This information can be found in the *Materials and Methods* section and the information of the HEK293 cells was now added to the figure legend.

2. Figures 1E, 3C: Please define your western blot labels (T, C, M).

Answer:

This information has been added to the respective figure legends (T, total; C, cytosol; M, mitochondria)

3. Figure 2M: Lane labeling 1-4 is not clear. Same labeling is found in other figures, but numbers are not specified.

Answer:

We consistently labeled the individual lanes in which sample was loaded with lane numbers. When needed, we referred to these numbers in the text.

4. Please correct the following to a more accurate statement: "We thereby observed only a minor but not significant stabilization of cytosolically confined AK2-3CS (Figure 3A)." Neither the western blot nor the graph shows a "minor stabilization of AK2".

Answer:

We corrected the statement. It now reads: We thereby observed **no significant** stabilization of cytosolically confined AK2-3CS (Figure 3A).

5. In figure 3A, AK2 stabilization experiments are done on a 3CS background. Then, a WT background was used for figure 3B and 3C. Please clarify what is the rationale for this change?

Answer:

The 3CS background was used, because mutating the cysteines in AK2 results in the cytosolic localization of the protein (Finger et al, EMBO J 2020). Cytosolically localized AK2 "reacts usually more pronounced" to changes of proteasomal or DPP activity. We later employed the P6W mutant also to test for its distribution between mitochondria and cytosol. This experiment can only be done with the WT protein that is capable of becoming imported into mitochondria.

6. Visual inspection of Figure 3C indicates an accumulation of AK2 in the mitochondrial fraction, but the authors solely focus on the cytoplasmic fraction. Please take this result into account.

Answer:

Indeed, the total levels of AK2 increase in the P6W mutant. The referee is correct in stating that mitochondrial levels strongly increase in the P6W mutant. We wanted to emphasize that the usual rapid cytosolic degradation of AK2 did not take place anymore leading to observable cytosolic amounts of AK2. Following the suggestion of the referee, we now adjusted the respective text part **(in red)**: In line with a decreased cytosolic degradation, AK2-P6W in part accumulated in the cytosolic fraction **although mitochondrial levels increased even more** (Figure 3C).

7. As shown in Figure 1E, AK2-HA WT is expected to be present in the mitochondrial fraction. However, in figure 3C this effect is no longer observed. How do the authors account for this discrepancy?

Answer:

We agree with this referee, that AK2-HA WT is expected to be present in the mitochondrial fraction and would like to emphasize that we also observe mitochondrial localization of AK2-HA WT in **Figure 3C** even though to a lower extent when compared to endogenous AK2 in **Figure 1E**. As we could faithfully demonstrate mitochondrial localization of AK2-HA in immunofluorescence and digitonin-fractionation experiments in our previous study (Finger*, Habich* et al., EMBO, 2020) we did not further investigate this discrepancy.

8. In figure 3A, what was used for normalization of HA signal?

Answer:

The signal was first normalized to the TCE signal and then plotted relative to the siScr control.

Dear Jan,

Thank you for the submission of your revised manuscript to EMBO reports. I have already forwarded you the referee reports and copy them again below.

As you will see, the referees find that the study has been significantly improved during revision and recommend publication after a few remaining concerns have been addressed.

From the editorial side, there are also a few things that we need before we can proceed with the official acceptance of your study:

- Your manuscript will be published in our Reports section, which requires the combination of Results & Discussion.
 - Please provide up to 5 keywords.
 - Regarding the Author Contributions, we now use CRediT to specify the contributions of each author in the journal submission system. Therefore, please remove the Author Contributions from the manuscript file and make sure that the author contributions in our online manuscript tracking system are correct and up-to-date. The information you specified in the system will be automatically retrieved and typeset into the article. You can enter additional information in the free text box provided, if you wish.
 - Please note that as per our editorial policies, all data supporting a statement need to be included in the manuscript. In this respect, we noted that the statement that AK2 is unstable and that only a small fraction resides in the cytosol is based on 'data not shown'. Do you have supporting evidence for this, even if it is a negative result? In this case, please include it, e.g., in the Appendix.
 - Please provide a complete author checklist, which you can download from our author guidelines (<<https://www.embopress.org/page/journal/14693178/authorguide>>). Please insert information in the checklist that is also reflected in the manuscript. The completed author checklist will also be part of the RPF.
 - We also need a Reagents and Tools table listing key reagents, experimental models, software and relevant equipment and including their sources and relevant identifiers.
Please download and fill our Reagents and Tools Table template (.docx), which you can find in our author guidelines: <https://www.embopress.org/page/journal/14693178/authorguide#structuredmethods>.
When submitting your revised manuscript, please do not include the Reagents and Tools Table in the Methods section of the manuscript but upload it as a separate file choosing the file type "Reagent Table".
An example of a Method paper with Structured Methods can be found here: <https://www.embopress.org/doi/10.15252/msb.20178071>.
 - Materials and Methods should be Methods
 - Data availability section:
It should only be used to refer to deposited datasets. Therefore, please remove the following statement: "The datasets generated during and/or analysed during the current study are available from the corresponding author on reasonable request." Please also include the URL that directly resolves to the dataset and remove the reviewer access, which we had included for the referees.
 - Appendix: please change the nomenclature to Appendix Fig. S# and Appendix Table S# (and Appendix references) throughout the file and the manuscript callouts.
 - Tables S1 - S6 could (should) be part of the Reagents and Tools table (see above). If you prefer to keep them separate please add callouts to these tables in the manuscript.
 - Figure S2B lacks a definition of "n", or does the description for panel C, n=3 biological replicates apply here too?
 - Figure S3, S4C: even though 'n' is defined, please also specify whether these are technical or biological replicates.
 - Our production/data editors have asked you to clarify several points in the figure legends (see below). Please incorporate these changes in the manuscript and return the revised file with tracked changes with your final manuscript submission.
- A) Statistical test information. Only p-values that are actually shown in the figure panel(s) should (and must) be defined in the legends, all others should be removed from (or added to) the legend. Moreover, we ask for the specification of exact p-values:
- Please indicate the statistical test used for data analysis in the legends of figures 2E, F, G, H, I, J; 3D, E; 4F, G.

B) Replicates and error bars:

- Please note that information related to n is missing in the legends of figures 2E, F, G, H, I, J; 3D, E; 4F, G
- Please note that the error bars are not defined in the legend of figure 1D. "

- Finally, EMBO Reports papers are accompanied online by

A) a short (1-2 sentences) summary of the findings and their significance,

B) 2-3 bullet points highlighting key results and

C) a schematic summary figure that provides a sketch of the major findings (not a data image).

Please provide the summary figure as a separate file in PNG or JPG format at a size of 550x300-600 pixels (width x height).

Please note that the size is rather small and that text needs to be readable at the final size. Please send us this information along with the revised manuscript.

With kind regards,

Martina

=====

Referee #1:

Overall, the authors have resolved most of my concerns, resulting in a much improved manuscript that is now suitable for publication.

However, there are still a few critical issues that require attention prior publication.

1. In vitro ubiquitination assays: It is unclear from the methods section which peptide sequences were used in this assay. If these are short peptides, they likely do not contain lysine residues and, therefore, cannot undergo ubiquitination. If KK-biotin is added to their C-termini, it does not accurately reflect natural ubiquitination processes. XIAP may interact with these peptides and ubiquitinate the transplanted KK sequence, but this may not occur with a full-length protein, where lysine accessibility differs. Additionally, the assays lack a control with XIAP but no peptide, which would help confirm that the ubiquitination observed is not due to autoubiquitination of the bound XIAP itself. This point needs clarification before acceptance. Based on that, it should be decided whether to include this data and, if so, to discuss its limitations.
2. Figure 4K does not support the proposed regulation of EIF2A by DPP9 and IAPs and, instead, weakens the overall findings of the paper. First, the input data shows no changes in EIF2A protein levels in MG132, DPP9 KO, or siXIAP/BIRC2 cells, which contradicts the idea of DPP9/IAPs/proteasomal degradation of EIF2A. Given that there is x10 more DPP9-processed, unacetylated EIF2A protein (Figure 4D), it should show altered protein levels under the outlined settings. Second, the IP immunoblot is faint, and there is no control IP (such as not precipitating EIF2A). I suspect that the ubiquitin signal observed is due to background ubiquitinated proteins non-specifically binding to the beads, rather than originating from EIF2A. Additionally, siXIAP/BIRC2 should show reduced EIF2A ubiquitination in WT cells, with an even more pronounced reduction in DPP9 KO cells, but this is not observed. I recommend removing this data from the paper.
3. All bar graphs need to be supplemented with significance levels (p values)
4. Page 11: "Likewise, a peptide representing the acetylated MAP- and DPP8/9 processed EIF2A did also not coprecipitate XIAP, BIRC2, BIRC3, or BIRC6"- change "did also not" to "did not also".
5. The use of "(MYP)" is confusing, as it could be interpreted as referring to tyrosine ("Y"). It may be preferable to use "X" instead, as it does not represent an amino acid.
6. It may be helpful to replace "TCE" with "Ponceau" in the western blots, as "TCE" is sometimes also shown in IPs, where it would not be appropriate.
7. The use of "(NH2-SV...)" is confusing, as it may be interpreted as the end of the sentence. It might be clearer to replace with "(NH2-SV...)" instead.

Referee #3:

The authors have addressed my comments adequately, including additional supporting data.

Points that we addressed in response to editorial request

- We combined the *Results and Discussion* section
- We provide 5 keywords
- We removed the *author contribution* section
- We provide now a reference to *Finger et al EMBO J 2020* where we previously wrote “data not shown” in the main text
- We provide a filled-in *author checklist*
- We provide a *reagents and tools table* listing and incorporated information provided in Table S1-S6
- We renamed *Materials and Methods* to *Methods*
- We removed the first sentence from the *Data availability* section. We provide the URL for the PRIDE data.
- We corrected the nomenclature and now write *Appendix Fig.* and *Fig.* in the main text and the SI.
- We added the number of biological replicates to Appendix Figure S2B
- In S3 and S4C biological replicates are meant. We clarified this in the text
- We clarified the following points as per request from the production/data editor:
 - We added the test used for statistical analysis to the figure legends
 - We added the number of replicated and the meaning of error bars.

- We provide the following *findings summary and significance* statement:

Lapacz et al uncover a novel regulatory mechanism controlling AK2 stability in the cytosol, linking mitochondrial protein processing to proteasomal degradation. They reveal that cytosolic degradation of adenylate kinase 2 (AK2) and potentially 129 additional proteins is regulated by Inhibitors of Apoptosis (IAPs) through a newly exposed IAP-binding motif (IBM) following processing by DPP8/9 unless counteracted by NatA-dependent N-terminal acetylation

- We provide the following bullet points summarizing the results:
 - *DPP8/9 processing exposes an IBM at the N-terminus of AK2*
 - *IAP E3 ligases facilitate AK2 degradation unless counteracted by NatA-dependent N-terminal acetylation.*
 - *A genome-wide analysis identified 129 potential proteins with similar IBM exposure.*

- We provide the following sketch of the major findings:

Response to reviewer#1

1. *In vitro* ubiquitination assays: It is unclear from the methods section which peptide sequences were used in this assay. If these are short peptides, they likely do not contain lysine residues and, therefore, cannot undergo ubiquitination. If KK-biotin is added to their C-termini, it does not accurately reflect natural ubiquitination processes. XIAP may interact with these peptides and ubiquitinate the transplanted KK sequence, but this may not occur with a full-length protein, where lysine accessibility differs. Additionally, the assays lack a control with XIAP but no peptide, which would help confirm that the ubiquitination observed is not due to autoubiquitination of the bound XIAP itself. This point needs clarification before acceptance. Based on that, it should be decided whether to include this data and, if so, to discuss its limitations.

The utilized peptides were short peptides which all contain two additional lysine residues added as described in **Fig. 2D**. This is now also clarified in the *Methods* section. The major ubiquitin acceptor in these experiments is however streptavidin which harbors 16 lysine residues and which is coupled to the peptides in the course of the experiment. We noticed this when we originally set up these *in vitro* experiments (Mueller et al., *SciAdv*, 2021). The entire experiment is set-up to guarantee full loading of streptavidin beads with peptides to ensure comparability between experiments. We agree with the reviewer that this does not represent natural available lysine residues as in the full-length proteins (although in AK2 18 of the 19 lysine residues are exposed in the mature protein), but the approach serves the purpose to test **whether XIAP specifically targets the free Ser-starting peptide**.

Autoubiquitination is a valid concern which we had as well as XIAP can be active on itself. We optimized the washing conditions and performed at the beginning XIAP WBs on the *in vitro* ubiquitination assays to control it. We want to point out that potential residual autoubiquitination is internally controlled in these experiments. **Potential autoubiquitination of XIAP would be present in in each sample to the same extent regardless which peptide is given as a substrate**. The ubiquitination assays in Fig. 3I and 4I show dramatic increase of the Ub signal for the N-terminal free Ser-starting peptide compared to the other tested peptides which would not be the case if we would score for XIAP autoubiquitination only.

The purpose of the experiment was to show that XIAP specifically targets the free Ser-starting peptide which these *in vitro* ubiquitination assays clearly demonstrate. Additionally, E3 ligases are very active and there is no reason to believe that the exposed lysines in AK2 or EIF2A will not be able to accept ubiquitin. We hope that this clarifies the reviewers concerns and that he agrees that we do not need to discuss further limitations for our data interpretation. We also decided to keep the data in our manuscript.

2. Figure 4K does not support the proposed regulation of EIF2A by DPP9 and IAPs and, instead, weakens the overall findings of the paper. First, the input data shows no changes in EIF2A protein levels in MG132, DPP9 KO, or siXIAP/BIRC2 cells, which contradicts the idea of DPP9/IAPs/proteasomal degradation of EIF2A. Given that there is x10 more DPP9-processed, unacetylated EIF2A protein (Figure 4D), it should show altered protein levels under the outlined settings. Second, the IP immunoblot is faint, and there is no control IP (such as not precipitating EIF2A). I suspect that the ubiquitin signal observed is due to background ubiquitinated proteins non-specifically binding to the beads, rather than originating from EIF2A. Additionally, siXIAP/BIRC2 should show reduced EIF2A ubiquitination in WT cells, with an even more pronounced reduction in DPP9 KO cells, but this is not observed. I recommend removing this data from the paper.

As suggested by the referee, we removed the data (Fig. 4K) and its description in the text and the figure legends.

3. All bar graphs need to be supplemented with significance levels (p values)

We added the significance levels to all bar graphs. However, we would like to point out that for many experiments with N=3 biological replicates this statistical analysis is not necessarily accurate. We believe it is more correct and credible to show individual data points, which we already did in the previous version of the manuscript.

4. Page 11: "Likewise, a peptide representing the acetylated MAP- and DPP8/9 processed EIF2A did also not coprecipitate XIAP, BIRC2, BIRC3, or BIRC6"-
change "did also not" to "did not also".

Done

5. The use of "(MYP)" is confusing, as it could be interpreted as referring to tyrosine ("Y"). It may be preferable to use "X" instead, as it does not represent an amino acid.

As suggested by this referee, we replaced the "Y" by "X" in the respective figures and in the text.

6. It may be helpful to replace "TCE" with "Ponceau" in the western blots, as "TCE" is sometimes also shown in IPs, where it would not be appropriate.

"TCE" refers to 2,2,2-Trichloroethanol which is an in-gel staining technique that is different from Ponceau. Thus, the term TCE is correct.

7. The use of "NH2-SV..." is confusing, as it may be interpreted as the end of the sentence. It might be clearer to replace with "(NH2-SV...)" instead.

We thank the referee for this comment, but respectfully disagree. We went through the text and found all peptide sequences well understandable and not to be confused with the end of sentences.

Prof. Jan Riemer
University of Cologne
Biochemistry
Zuelpicher Str. 47a
Cologne 50674
Germany

Dear Prof. Riemer,

I am very pleased to accept your manuscript for publication in the next available issue of EMBO reports. Thank you for your contribution to our journal.

Kind regards,
